https://doi.org/10.1038/s41467-021-23199-5　　**OPEN**

# Functional and structural characterization of a two-MAb cocktail for delayed treatment of enterovirus D68 infections

Chao Zhang 1,2,5, Cong Xu3,5, Wenlong Dai2,5, Yifan Wang3, Zhi Liu2, Xueyang Zhang2, Xuesong Wang2, Haikun Wang 2, Sitang Gong 1✉, Yao Cong 3,4✉ & Zhong Huang 2✉

Enterovirus D68 (EV-D68) is an emerging pathogen associated with respiratory diseases and/or acute flaccid myelitis. Here, two MAbs, 2H12 and 8F12, raised against EV-D68 virus-like particle (VLP), show distinct preference in binding VLP and virion and in neutralizing different EV-D68 strains. A combination of 2H12 and 8F12 exhibits balanced and potent neutralization effects and confers broader protection in mice than single MAbs when given at onset of symptoms. Cryo-EM structures of EV-D68 virion complexed with 2H12 or 8F12 show that both antibodies bind to the canyon region of the virion, creating steric hindrance for sialic acid receptor binding. Additionally, 2H12 binding can impair virion integrity and trigger premature viral uncoating. We also capture an uncoating intermediate induced by 2H12 binding, not previously described for picornaviruses. Our study elucidates the structural basis and neutralizing mechanisms of the 2H12 and 8F12 MAbs and supports further development of the 2H12/8F12 cocktail as a broad-spectrum therapeutic agent against EV-D68 infections in humans.

1 Joint Center for Infection and Immunity, Guangzhou Institute of Pediatrics, Department of Gastroenterology, Guangzhou Women and Children's Medical Center, Guangzhou Medical University, Guangzhou, China. 2 CAS Key Laboratory of Molecular Virology & Immunology, Institut Pasteur of Shanghai, Chinese Academy of Sciences, University of Chinese Academy of Sciences, Shanghai, China. 3 State Key Laboratory of Molecular Biology, National Center for Protein Science Shanghai, Shanghai Institute of Biochemistry and Cell Biology, Center for Excellence in Molecular Cell Science, Chinese Academy of Sciences, University of Chinese Academy of Sciences, Shanghai, China. 4 Shanghai Science Research Center, Chinese Academy of Sciences, Shanghai, China. 5These authors contributed equally: Chao Zhang, Cong Xu, Wenlong Dai. ✉email: sitangg@126.com; cong@sibcb.ac.cn; huangzhong@ips.ac.cn

Enterovirus D68 (EV-D68) is a small non-enveloped virus belonging to the D species of the *Enterovirus* genus within the *Picornaviridae* family[1]. The prototype strain of EV-D68, Fermon, was originally isolated from pediatric patients with pneumonia and bronchiolitis in the United States in 1962[2]. Except the strain Fermon, all other EV-D68 strains can be divided into four primary clades, namely A, B, C, and D (previously described as A2) based on VP1 nucleotide sequence[3,4]. EV-D68 infection can cause acute respiratory illness and/or severe neurological disorder mainly acute flaccid myelitis (AFM) in children[5–7]. Over the past decade, EV-D68 has become widespread all over the world and continues to cause both outbreaks and sporadic cases[8]. In particular, from August 2014 to January 2015, a nationwide outbreak of EV-D68 infection occurred in the United States, resulting in 1153 confirmed cases including 14 deaths[8,9]. EV-D68 outbreaks were also witnessed in the United States in 2016 and in 2018[10,11], coinciding with the occurrence of 153 and 237 AFM cases, respectively[12]. In addition, an upsurge of EV-D68 infection was reported in several European countries in 2016[13–15], accompanied by the identification of 29 EV-D68-associated AFM cases[16]. Clearly, EV-D68 has become a serious global health concern.

Like other enteroviruses, EV-D68 possesses a ~30 nm icosahedral capsid composed of 60 protomers, each consisting of VP1, VP2, VP3, and VP4 subunit proteins[17]. The main structural features of EV-D68 capsid include star-shaped mesa at the five-fold axis, narrow depression (the canyon) around each mesa, VP1 hydrophobic pocket directly beneath the canyon floor, and prominent three-bladed propeller at the three-fold axis[17–19]. Two distinct cellular receptors for EV-D68 have been identified: sialic acid and neuron-specific intercellular adhesion molecule-5 (ICAM-5/telencephalin)[20–22]. Sialic acid has been reported to bind into the virus canyon[21], whereas binding site of ICAM-5 is still unknown. Binding of cellular receptors or treatment with acid trigger a series of conformational changes in the virus, resulting in two expanded uncoating intermediates called the expanded 1 (E1) and A (altered; 135S) particle[18,19,21]. The E1 particle is a newly identified expanded state exhibiting a majority of internal regions (VP4 and VP1 N terminus) to be ordered and serves as an intermediate in transition from native mature virion to A-particle[18]. For A-particle, the N-terminal tail of VP1 is externalized and VP4 is expelled from the capsid[18]. Subsequently, viral RNA is released from the A particles into the cytoplasm, leaving behind an empty capsid shell (termed the 80S particle).

EV-D68 poses a major global threat to children's health; however, neither vaccine nor therapeutic agent for EV-D68 is currently available. Monoclonal antibodies (MAb) are a viable option for developing antiviral drugs, as demonstrated by the successful commercialization of palivizumab, a humanized MAb against respiratory syncytial virus[23]. Thus far, very limited efforts have been made towards developing therapeutic MAbs for treating EV-D68 infections[19,24,25].

Here, we show the discovery and structural characterization of a two-MAb cocktail for delayed treatment of EV-D68 infections. We isolate two EV-D68-specific neutralizing MAbs, 2H12 and 8F12, from mice immunized with recombinant EV-D68 virus-like particles (VLP)[26,27]. MAb 2H12 differs significantly from 8F12 in their antigen-binding and neutralization profiles, yet they complement each other in conferring broad-spectrum neutralization in vitro and cross-clade protection in vivo. Notably, the 2H12/8F12 antibody cocktail is able to effectively treat EV-D68-infected mice even when administered three days after viral challenge, a time point by which the virus has spread into the brain and spinal cord[28]. The high resolution (up to 2.9 Å) cryo-electron microscopy (cryo-EM) structures of EV-D68 virion in complex with 2H12 or 8F12 antigen-binding fragments (Fab) show that both

MAbs target the same previously unreported antigenic site located at the south rim of the canyon yet they exert distinct effects on EV-D68 virion stability, highlighting the uniqueness of this pair of MAbs. Our structural analyses also identify a new uncoating intermediate state (designated S3 in this study) which has not been previously reported for picornaviruses. At last, we demonstrate that human-mouse chimeric versions of 2H12 and 8F12 retain neutralization potency and therapeutic efficacies. Our work thus lays a solid foundation for further development of the 2H12/8F12 cocktail into a pan-EV-D68 therapy, which remains an urgent unmet medical need.

## Results

**Generation and biochemical characterization of anti-EV-D68 neutralizing MAbs.** It has been previously shown that recombinant VLPs of EV-D68 could induce high levels of broadly neutralizing antibodies capable of conferring protection against lethal infection in mice[26,27]. In this study, we adopted the hybridoma technology to generate EV-D68-specific neutralizing MAbs from one mouse immunized with recombinant VLP of EV-D68 strain US/MO/14-18950 (hereinafter referred to as 18950, clade B). The resulting hybridoma cells were screened for their ability to neutralize EV-D68 clinical strain US/MO/14-18947 (hereinafter referred to as 18947; clade B). Note that strain 18947 was used in the initial screening because the immunogen strain (18950) was not available to us. A summary of all the EV-D68 strains used in this study is presented in Supplementary Table 1. Two stable clones (2H12 and 8F12) were found to possess neutralizing activity (Fig. 1a). Isotyping analysis showed that 2H12 was IgG2a antibody while clone 8F12 was of IgG2b isotype (Fig. 1a). We determined the coding sequences of the two anti-EV-D68 MAbs. The sequence of the heavy chain variable region ($V_H$) of antibody 2H12 was 79% identical to that of 8F12, while the two MAbs shared 57% sequence identity in light-chain variable region ($V_L$) (Supplementary Fig. 1). Next, neutralization potency of the purified MAbs was initially assessed by standard neutralization assay with the 18947 strain. For MAbs 2H12 and 8F12, their neutralization concentrations (the lowest antibody concentration that could fully inhibit EV-D68-induced cytopathic effect [CPE]) against strain 18947 were determined to be 1.95 and 0.06 μg/mL, respectively (Fig. 1a). In quantitative neutralization assays, both anti-EV-D68 MAbs exhibited inhibitory effects in an antibody dose-dependent manner (Fig. 1b) with IC50s being 0.412 and 0.004 μg/mL for 2H12 and 8F12, respectively; in contrast, the two irrelevant MAbs, 1C11 (IgG2a isotype control) and 1F4 (IgG2b isotype control) did not show any neutralization effect regardless of the antibody dose.

The MAbs were tested by ELISA for their ability to recognize different antigens, including EV-D68 VLP, enterovirus 71 (EV71) VLP, and coxsackievirus A16 (CVA16) VLP. Anti-EV-D68 MAbs 2H12 and 8F12, but not the isotype control antibodies 1C11 and 1F4, were found to react with EV-D68 VLP (Fig. 1c). It is worth pointing out that MAb 2H12 showed stronger binding to EV-D68 VLP than MAb 8F12 (Fig. 1c). The two anti-EV-D68 MAbs did not exhibit any reactivity in EV71 VLP- or CVA16 VLP-binding ELISA assays which were validated using anti-EV71 MAb D5 and anti-CVA16 MAb 9B5, respectively (Supplementary Fig. 2). These results indicated that MAbs 2H12 and 8F12 were indeed EV-D68-specific antibodies.

Compared with MAb 2H12, MAb 8F12 had relatively lower binding activity towards EV-D68 VLP (Fig. 1c) but exhibited stronger neutralizing activities on EV-D68 strain 18947 (Fig. 1b). This seeming contradiction prompted us to determine the binding affinity of the MAbs to the 18947 virion by bio-layer interferometry (BLI). The virion-binding affinity of MAb 8F12 was found to be significantly higher than that of 2H12 (Fig. 1d, e),

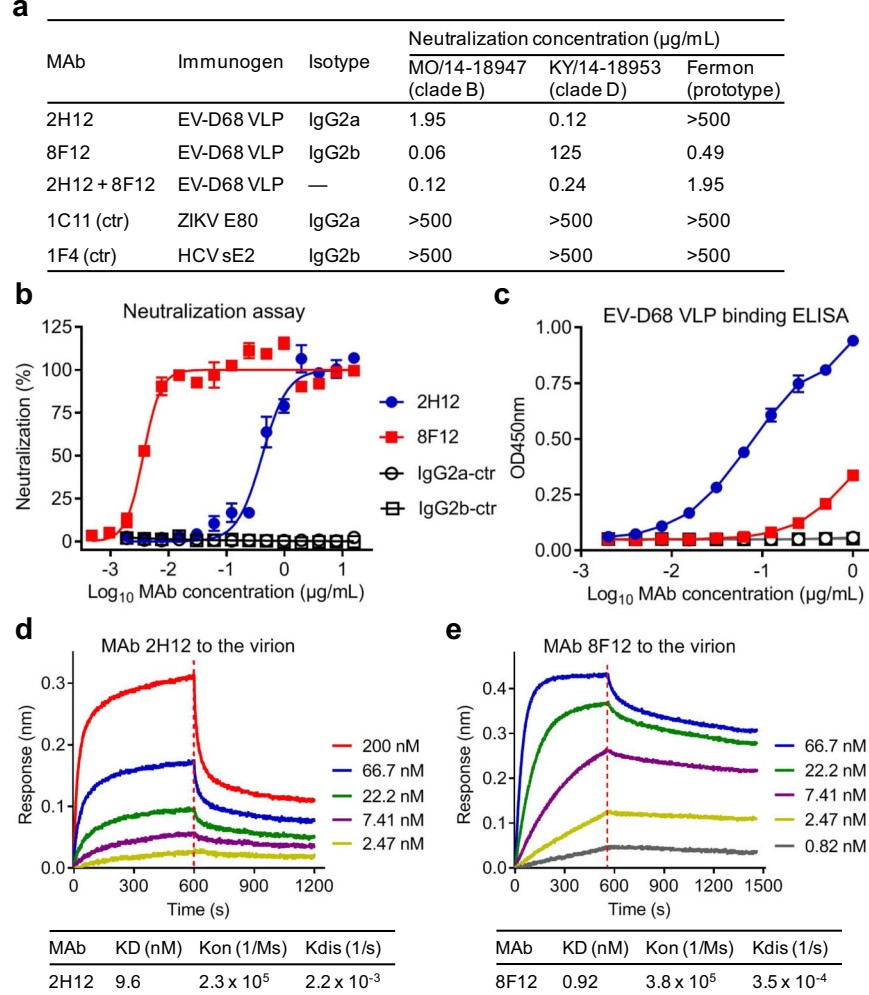

**Fig. 1 Neutralization activity and binding properties of the MAbs. a** Isotypes and neutralization activity of anti-EV-D68 MAbs. Neutralization concentrations of the MAbs were defined as the lowest antibody concentrations that completely prevented virus-induced cytopathic effect. 2H12+ 8F12, MAbs 2H12, and 8F12 were combined at a ratio of 1:1. ctr, isotype control. Symbol (—), not tested. **b** Neutralization activity of the MAbs against EV-D68. 100 $TCID_{50}$ of EV-D68 strain 18947 was incubated with two-fold serial dilutions of purified MAbs 2H12 and 8F12 for 1 h before adding to RD cells. Cell viability was measured by CellTiter-Glo 2.0 assay 3 days after infection. Data are expressed as mean ± standard error of mean (SEM) of triplicate wells. **c** Reactivities of the MAbs towards EV-D68 VLP determined by ELISA. Data are expressed as mean ± standard deviation (SD) of triplicate wells. In panels (**b**, **c**), ZIKV-specific MAb 1C11 and HCV-specific MAb 1F4 served as IgG2a and IgG2b isotype controls (ctr), respectively. **d, e** Binding affinities of the MAbs to EV-D68 18947 virion measured by BLI. Association and dissociation steps are divided by dotted red line. MAb concentrations used and values of KD, Kon and Kdis were shown.

in agreement with the neutralization potency of the MAbs (Fig. 1b). Hence, for the anti-EV-D68 MAbs, their virion-binding affinity, rather than the affinity towards the VLP immunogen, positively correlated with their neutralization ability.

**Cross-neutralization capacities of the MAbs**. The MAbs were further evaluated for their ability to cross-neutralize two other EV-D68 strains, including the prototype strain Fermon and a clinical strain US/KY/14-18953 (hereinafter referred to as 18953; clade D). MAb 2H12 effectively neutralized strain 18953 with a neutralization concentration of 0.12 μg/mL but had no neutralization effects on Fermon even at 500 μg/mL, the highest concentration tested (Fig. 1a). Conversely, MAb 8F12 showed very weak neutralization against strain 18953 (neutralization concentration: 125 μg/mL), but it could potently neutralize the Fermon strain (neutralization concentration: 0.49 μg/mL). As expected, the two isotype control antibodies, 1C11 and 1F4, did not display any neutralization effect on the two EV-D68 strains (Fig. 1a).

The distinct cross-neutralization profiles of 2H12 and 8F12 propelled us to investigate whether they can be used in combination to improve neutralization breadth. A two-MAb cocktail was thus formulated by mixing 2H12 and 8F12 at a ratio of 1:1. Results from neutralization assays showed that the two-MAb cocktail exhibited greater neutralization breadth than any of single MAbs (Fig. 1a). Specifically, for the antibody cocktail, neutralization concentrations against strains 18947, 18953, and Fermon were determined to be 0.12, 0.24, and 1.95 μg/mL, respectively (Fig. 1a). These data indicated that 2H12 and 8F12 MAbs can complement each other in neutralizing diverse EV-D68 clades/strains. In another word, neutralization breadth was improved by combining the 2H12 and 8F12 antibodies, providing a strong rationale for using the cocktail but not single MAbs for pan-EV-D68 neutralization.

**Prophylactic efficacy of the MAbs**. The protective efficacy of anti-EV-D68 MAbs was evaluated in a previously established

mouse model of EV-D68 infection[28]. For prophylactic efficacy evaluation, five groups of newborn ICR mice ($n = 12$–14/group) were given PBS, or a single dose (10 μg/g) of 2H12, 8F12, IgG2a isotype control (1C11), or IgG2b isotype control (1F4), respectively. One day later, all mice were infected with EV-D68 clade B strain 18947 and subsequently observed for clinical signs and mortality for a period of 14 days. As shown in Supplementary Fig. 3, mice in the PBS and control antibody groups started to show clinical signs at 3 days post infection (dpi), and majority of them eventually died (69% and 77% mortality, respectively). In contrast, 92% and 100% of the mice in the 2H12 and 8F12 groups survived from lethal challenge, respectively. These results indicated that both 2H12 and 8F12 MAbs are effective in preventing lethal EV-D68 infection in mice.

**Therapeutic efficacy of the MAbs.** We then assessed the therapeutic efficacy of the MAbs in the mouse model. The first experiment was designed to evaluate therapeutic efficacies of the MAbs administered at 1 dpi. Groups of 1-day-old ICR mice were infected with strain 18947 and 1 day later given single injections of PBS, 2H12 (10 μg/g), or 8F12 (10 μg/g), respectively. Survival and clinical score were then monitored on a daily basis. As shown in Fig. 2a, mice in the PBS group became sick at 3 dpi and a large proportion (86%) of them eventually died; in contrast, all of the MAb-treated mice survived, indicating that the MAbs were therapeutically efficacious. We should point out that the mean clinical scores in the 2H12 group were significantly higher than those in the 8F12 groups (Fig. 2a).

The second experiment was aimed to evaluate the MAbs for their efficacy in treating mice that had already developed disease. Besides individual MAbs, a two-MAb cocktail (formulated by mixing 2H12 and 8F12 at a ratio of 1:1) was also included as a treatment option in the experiments. Mice infected with strain 18947 started to show clinical signs at 3 dpi (Fig. 2a, right panel). Therefore, MAb treatment was given at 3 dpi and the treated mice were monitored for 2 weeks. As shown in Fig. 2b, the survival rates for the 2H12, 8F12, and two-MAb groups were 85%, 92%, and 92%, respectively, whereas only 26% of the mice in the PBS control group survived. These data showed that the anti-EV-D68 MAb treatment remained effective when given at a delayed time point during the course of infection. Again, clinical manifestations shown in the 2H12-treated mice were more severe than those observed in the 8F12 and the cocktail groups (Fig. 2b, right panel), suggesting that 2H12 is relatively less efficient than 8F12 in protecting against 18947 infection, in agreement with the lower in vitro neutralization efficiency of 2H12 against 18947 strain (Fig. 1b).

In a following experiment, we evaluated the anti-EV-D68 MAbs for their therapeutic breadth by using a clade D strain, 18953, as the challenge virus. Mice were administered single doses of MAb 2H12 (10 μg/g), MAb 8F12 (10 μg/g), the antibody cocktail (10 μg/g of 2H12 plus 10 μg/g of 8F12), or PBS 3 days after 18953 infection and subsequently monitored for a period of 14 days. The results were shown in Fig. 2c. In the control (PBS) group, the disease severity gradually increased and the final mortality rate reached 38%. The survival rate and mean clinical scores of the 8F12 group were not significantly different ($P$ value = 0.30) from those of the control group, suggesting that MAb 8F12, at the dose used, is not therapeutically effective against 18953 infection. In contrast, all of the mice receiving MAb 2H12 eventually recovered (100% survival), indicating that 2H12 is more effective than 8F12 in treating 18953 infection. Moreover, treatment with the 2H12/8F12 cocktail also led to full protection. Taken together, the above results demonstrate that the 2H12/8F12 cocktail is very effective in treating diverse EV-D68 infections at a delayed time point.

**Mode of action of 2H12 and 8F12 antibodies.** To elucidate the working mechanism of the anti-EV-D68 MAbs, we firstly determined at which stage the MAbs exert inhibitory function by performing time-of-addition assays. MAbs 2H12 and 8F12 were separately mixed with EV-D68 strain 18947 before virus binding to cells at 4 °C (pre-attachment) or added to virus-bound cells that had been incubated at 33 °C for various times, such as 0, 0.5, 1, 2, or 5 h (post attachment). The samples were collected at 8 h post infection and then subjected to RNA extraction and quantitative RT-PCR analysis. As shown in Fig. 3a, pretreatment with 2H12 or 8F12 almost completely abolished viral infection, while the MAbs were partially effective when added right before virus-bound cells were transferred to 33 °C (0 h post infection). In contrast, no inhibitory effect was seen when the MAbs were added at delayed time points (0.5–5 h post infection) (Fig. 3a). These results indicate that both 2H12 and 8F12 exert inhibition primarily at the pre-attachment stage.

Next, we performed attachment-inhibition assays to evaluate the effect of MAb treatment on virus attachment. Pre-incubation of the 18953 virus with 2H12 MAb reduced the amount of virus adsorbed onto the cells in an antibody dose-dependent manner whereas the isotype control antibody 1C11 had no inhibitory effect regardless of the antibody dose (Fig. 3b). Similarly, 8F12 pretreatment dose-dependently inhibited the 18947 virus attachment onto cells whereas the corresponding isotype control antibody 1F4 did not show any inhibition (Fig. 3c). These results demonstrate that both 2H12 and 8F12 antibodies can efficiently block EV-D68 attachment onto target cells.

Sialic acid is a receptor for EV-D68 that promotes viral attachment onto host cells[21]. It has been reported that EV-D68 can agglutinate red blood cells (RBC) through their sialic acid moiety[29]. We, therefore, performed hemagglutination inhibition (HI) assay to determine whether MAb treatment could interfere with the interaction between EV-D68 virion and sialic acid receptor. As shown in Fig. 3d, RBC alone sank to the bottom of the plate and formed red dots in the center of the wells, while the addition of EV-D68 18947 virion alone caused complete hemagglutination of RBC as indicated by the formation of red sheets across the wells. Pre-incubation of EV-D68 with the isotype control MAbs did not show any inhibition on hemagglutination regardless of the antibody dose. In contrast, MAbs 2H12 or 8F12 was found to exert HI in an antibody dose-dependent manner. It is worth noting that MAb 8F12 showed greater HI activity than MAb 2H12, in line with the higher neutralization efficiency of 8F12 against the 18947 strain (Fig. 1b). These results suggest that both 2H12 and 8F12 antibodies can inhibit the binding of EV-D68 to sialic acid receptor.

**Cryo-EM structure of EV-D68 in complex with 8F12 Fab.** To investigate the molecular basis of EV-D68 neutralization by the 8F12 MAb, we determined the cryo-EM structure of EV-D68 mature virion in complex with 8F12 Fab to a nominal resolution of 2.89 Å (Fig. 4a, Supplementary Fig. 4). Inspection of the original micrographs of the EV-D68/8F12 Fab (abbreviated as EV-D68/8F12) complex indicated a high occupancy of the Fab on the capsid with a hedgehog-like appearance (distinct from the smooth spherical appearance of EV-D68 virion[18,28]) (Supplementary Fig. 4a). We then built an atomic model for the EV-D68/8F12 complex (Fig. 4b, c), which matches the corresponding density map very well (Supplementary Fig. 5). Most of the side-chains can be well resolved in our map, indicating the high resolution of our map. We also show the central section of the EV-D68/8F12 map to illustrate the quality of the densities corresponding to the viral capsid and Fab (Supplementary Fig. 4e). Here only models of the variable regions of the Fab were built,

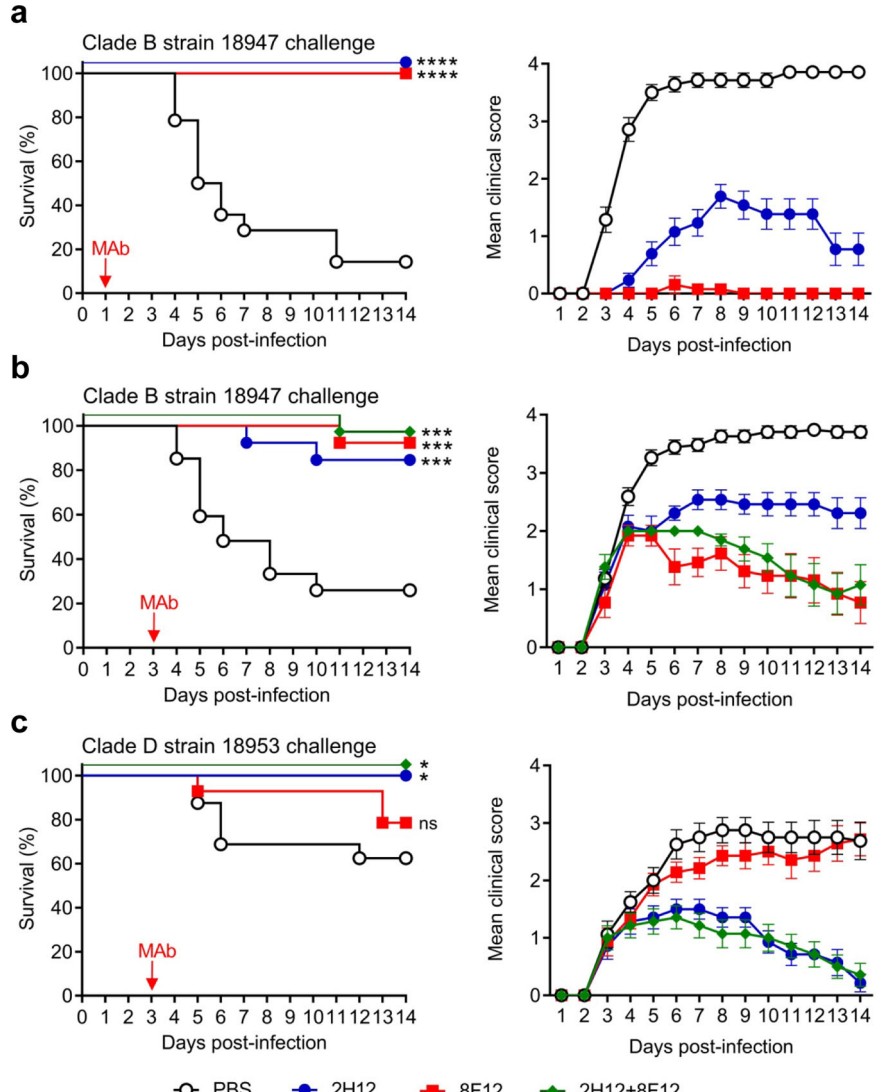

**Fig. 2 Therapeutic efficacy of the MAbs against EV-D68 infection in mice.** One-day-old ICR mice ($n = 13$–$27$/group) were inoculated intraperitoneally (i.p.) with strain 18947 (**a**–**b**) or strain 18953 (**c**). The suckling mice were i.p. injected with PBS, 10 μg/g of 2H12, 10 μg/g of 8F12, or a mixture of both MAbs (10 μg/g of each MAb) at 1 day post infection (dpi) (**a**) or 3 dpi (**b**, **c**) and were then monitored daily for survival and clinical score. Red arrows indicate the time points of MAb administration. Clinical scores were graded as follows: 0, healthy; 1, lethargy and reduced mobility; 2, limb weakness; 3, limb paralysis; 4, death. Note that to prevent overlap, the overlapping data sets in left panels (survival curves) were nudged by 5 units in the Y direction. Survival rates of antibody-treated mice were compared with the mice in the PBS control group. Statistical significance was determined by Log-rank (Mantel-Cox) test and was indicated as follows: ns., no significant difference ($p \geq 0.05$); *, $p < 0.05$; ***, $p < 0.001$; ****, $p < 0.0001$. In panel (**a**), $p$ value between the 2H12 or 8F12 group and the PBS control group is below 0.0001. In panel (**b**), $p$ value between the 8F12 or 2H12+8F12 group and the PBS group is 0.0001; $p$ value between the 2H12 group and the PBS group is 0.0007. In panel (**c**), $p$ value between the 2H12 or 2H12+8F12 group and the PBS group is 0.0121; $p$ value between the 8F12 group and the PBS group is 0.2958. All error bars represent SEM.

since electron densities corresponding to the constant regions were relatively weak (Supplementary Fig. 4d, more details for model building were described in the Methods).

The Fabs bind to the tips of the three-bladed propeller-like protrusion of EV-D68 and face each other across the icosahedral two-fold axis (Fig. 4a). The capsid structure in the EV-D68/8F12 complex is essentially identical to that of the unbound native EV-D68 virion[18] with the overall Cα root-mean-square deviation (RMSD) value being rather small (0.53 Å) between the protomers of the two structures (Supplementary Table 3); also, the two-fold axis channel remains closed in the complex (Fig. 4c). Collectively, these data indicate that 8F12 Fab binding does not induce obvious conformational changes of the EV-D68 virion. According to the structure, each protomer of EV-D68 engages with an 8F12 Fab

(Fig. 4d). 8F12 Fab binds to the south rim of the canyon, and its light chain is tilted towards the five-fold axis of the virus, obscuring the canyon region (Fig. 4d, f), which is involved in receptor binding[21]. Furthermore, the 8F12-binding footprints overlap with the sialic acid binding site (Fig. 4f), thus explaining the experimental observation that MAb 8F12 could block virus attachment to the sialic acid receptor (Fig. 3d).

To define antibody-binding epitopes, we analyzed the interactions between 8F12 Fab and EV-D68 virion. This analysis suggested that the heavy chain (specifically only complementary determining region [CDR] 1) of 8F12 Fab binds to the EF loop of VP2 (Fig. 4e①), while the light chain (framework [FR] 1, CDR1, FR3, and CDR3) of 8F12 interacts with the VP1 GH and BC loops, VP2 EF loop, and VP3 C-terminus of EV-D68 (Fig. 4e,

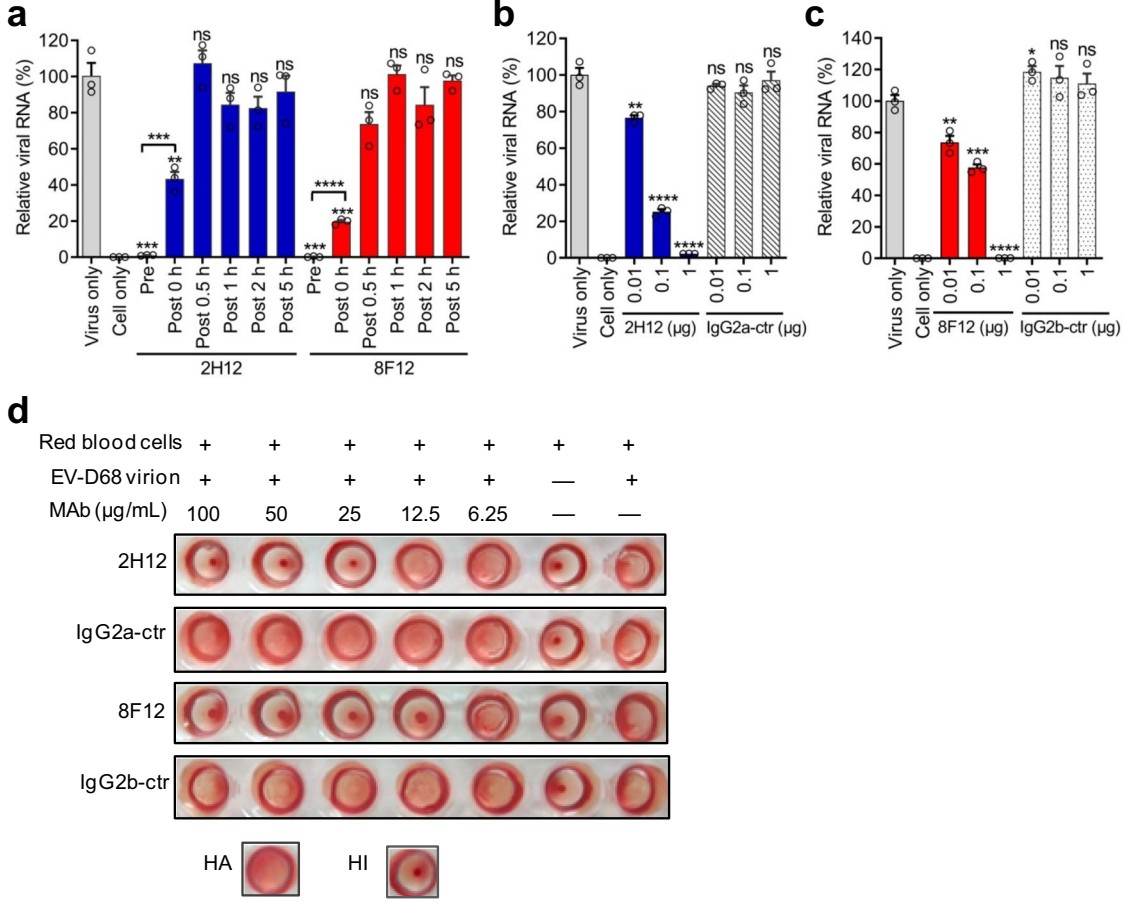

**Fig. 3 Anti-EV-D68 MAbs inhibited viral attachment and infection of cells. a** Pre- and post-attachment inhibition assays. 1000 $TCID_{50}$ of strain 18947 was exposed to the MAbs (2H12 or 8F12) before (pre-attachment, [Pre]) or at different time points after (post-attachment, [Post]) the virus was allowed to adsorb to cooled cells. Total RNA was isolated at 8 h after infection, and EV-D68 RNA was determined by real-time RT-PCR. **b, c** Inhibition of virus attachment by MAbs 2H12 and 8F12. $1.0 \times 10^7$ $TCID_{50}$ of strains 18953 (**b**) or 18947 (**c**) were pre-incubated with various amounts of the indicated antibodies (anti-EV-D68 MAbs 2H12 and 8F12, or isotype control MAbs) for 1 h prior to binding to prechilled RD cells at 4 °C for 1 h. After washing, cells were harvested and total RNA was isolated for real-time RT-PCR analysis of RNA contents of cell-bound virus. For each treatment in panels (**a–c**), viral RNA levels relative to those for the only virus-infected group are shown. Data are mean ± SEM of triplicate wells. Each symbol represents one well. Statistical significance between virus-only and treated groups was determined by a two-tailed Student's *t*-test and was indicated as follows: ns, no significant difference ($p \geq 0.05$); *, $p < 0.05$; **, $p < 0.01$; ***, $p < 0.001$; ****, $p < 0.0001$. In panel (**a**), $p$ value between the Pre groups and the virus-only group is 0.0001. In panel (**b**), for the 0.01-μg 2H12 group, $p = 0.0042$. In panel (**c**), for the 0.01-μg 8F12 group, $p = 0.0084$; for the 0.1-μg 8F12 group, $p = 0.0005$. **d** Dose-dependent hemagglutination inhibition (HI) by anti-EV-D68 MAbs 2H12 and 8F12. The assay relies on hemagglutination (HA) activity of EV-D68 virion (strain 18947). Symbol (+) indicates presence; (−) indicates absence. In Panels (**b–d**), ZIKV-specific MAb 1C11 and HCV-specific MAb 1F4 were used as IgG2a and IgG2b isotype controls (ctr), respectively.

Supplementary Table 4). Note that VP1 BC loop resides at the north rim of the canyon, while VP1 GH loop and VP2 EF loop are situated at the south rim (Fig. 4d, e). The 8F12/EV-D68 interaction interface covers ~1130 Å$^2$ of the surface area on each protomer, and the heavy and light chains of 8F12 contribute 30.4% and 69.6% of the interaction interface, respectively (Supplementary Table 6).

**Cryo-EM structures of EV-D68 in complex with 2H12 Fab reveal 2H12-induced particle transformation.** To reveal the structural basis of 2H12-mediated neutralization of EV-D68, we also performed cryo-EM study of the EV-D68 mature virion in complex with the 2H12 Fab. Surprisingly, the original cryo-EM micrographs of the EV-D68/2H12 Fab complex (abbreviated as EV-D68/2H12) suggested a small proportion (~15%) of the 2H12 Fab-bound capsids were broken into pieces, leaking genomic RNAs into the external fluid surrounding the particles (Supplementary Fig. 6a). Note that the two immune complexes, EV-D68

complexed with 8F12 or 2H12, were prepared at the same time using the same batch of EV-D68 antigen. Thus, it is very likely that 2H12, but not 8F12, could to some extent destroy EV-D68 viral particles.

Interestingly, from the same EV-D68/2H12 dataset we obtained three distinct conformational states, namely S1, S2, and S3, at resolutions of 3.09 Å, 3.60 Å, and 3.60 Å, respectively (Fig. 5a–c, Supplementary Fig. 6), which allowed us to build an atomic model for each of the three states and the models match the corresponding maps very well (Fig. 5d–f, Supplementary Fig. 7). Among these states, S1 occupies the highest population distribution (43.8%), while S2 and S3 occupy 22.9% and 33.3% of the population, respectively (Supplementary Fig. 6a). The overall structure of the S1 state of EV-D68/2H12 is similar to that of 8F12 Fab-bound EV-D68 virion (Figs. 4a and 5a), suggesting that 2H12 and 8F12 Fabs may target the similar region of the EV-D68 capsid. The S2 and S3 states largely resemble S1 in their external appearance (Fig. 5b, c); however, detailed examination revealed differences in their capsid structural features (Fig. 5d–f).

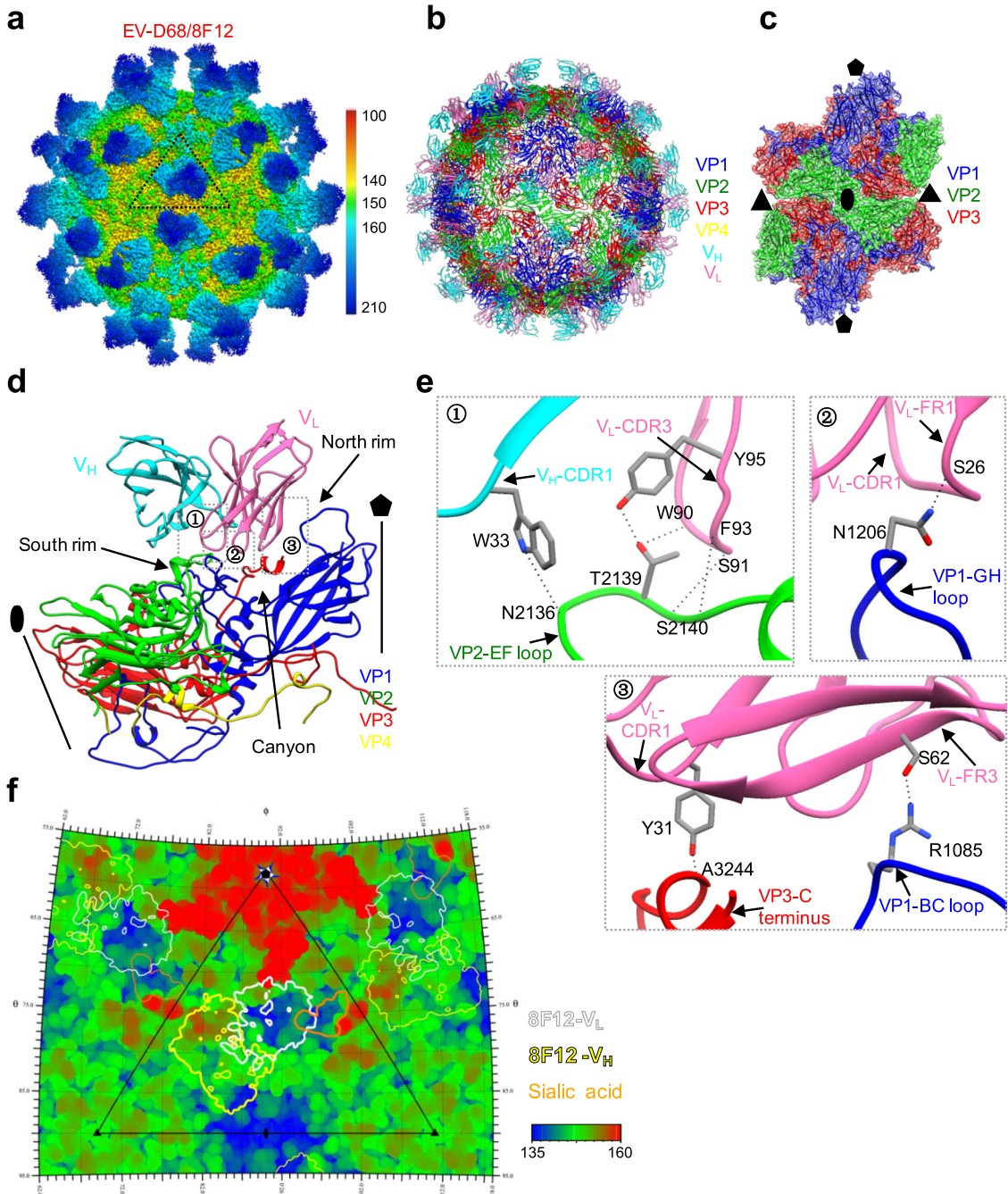

**Fig. 4 Cryo-EM structure of EV-D68 in complex with 8F12 Fab. a** Cryo-EM density map of EV-D68/8F12 complex viewed along the two-fold axis. The color bar indicates the corresponding radius from the center of the particle (unit in Å). The black triangle indicates one icosahedral asymmetric unit. **b** Atomic model of EV-D68/8F12 complex viewed along the two-fold axis. 8F12-$V_H$, 8F12-$V_L$, VP1, VP2, VP3, and VP4 are colored in cyan, hot pink, blue, green, red, and yellow, respectively. Models for only the variable regions of 8F12 Fab were built. **c** Density map with fitted models of four adjacent protomers around the two-fold axis. 8F12 Fab was removed for clarity. The black pentagon, triangle, and ellipse represent the five-fold, three-fold, and two-fold axes, respectively. **d** Binding Interface between EV-D68 protomer and 8F12 Fab. The canyon, north rim, and south rim are indicated by black arrows. The five-fold axis is also shown. **e** Zoom-in views of the three dotted boxed region in Panel (**d**), showing interactions between VP2 EF loop, VP1 BC, GH loops, and VP3 C-terminus of EV-D68 and the CDR and framework (FR) regions of 8F12 Fab. The images were rotated by various angles in order to better show the interactions. Possible hydrogen bonds in the interaction interface are indicated by black dashed lines. The residues of viral proteins are identified: the first digit represents viral capsid protein 1, 2, or 3, and the next three digits indicate position from the N-terminal. **f** Roadmap illustrating the footprints of 8F12 Fab on the EV-D68 virion surface, generated using RIVEM (Radial Interpretation of Viral Electron density Maps; reference[62]). Polar angles θ and φ represent latitude and longitude, respectively. Viral residues are colored according to the radius. The color bar indicates the corresponding radius (unit in Å). The $V_L$ and $V_H$ are indicated by white and yellow contour lines, respectively. Footprint of the sialic acid receptor (generated using EV-D68-6'SLN model, PDB: 5BNO) is also shown in orange. The icosahedral asymmetric unit is indicated by a black triangle.

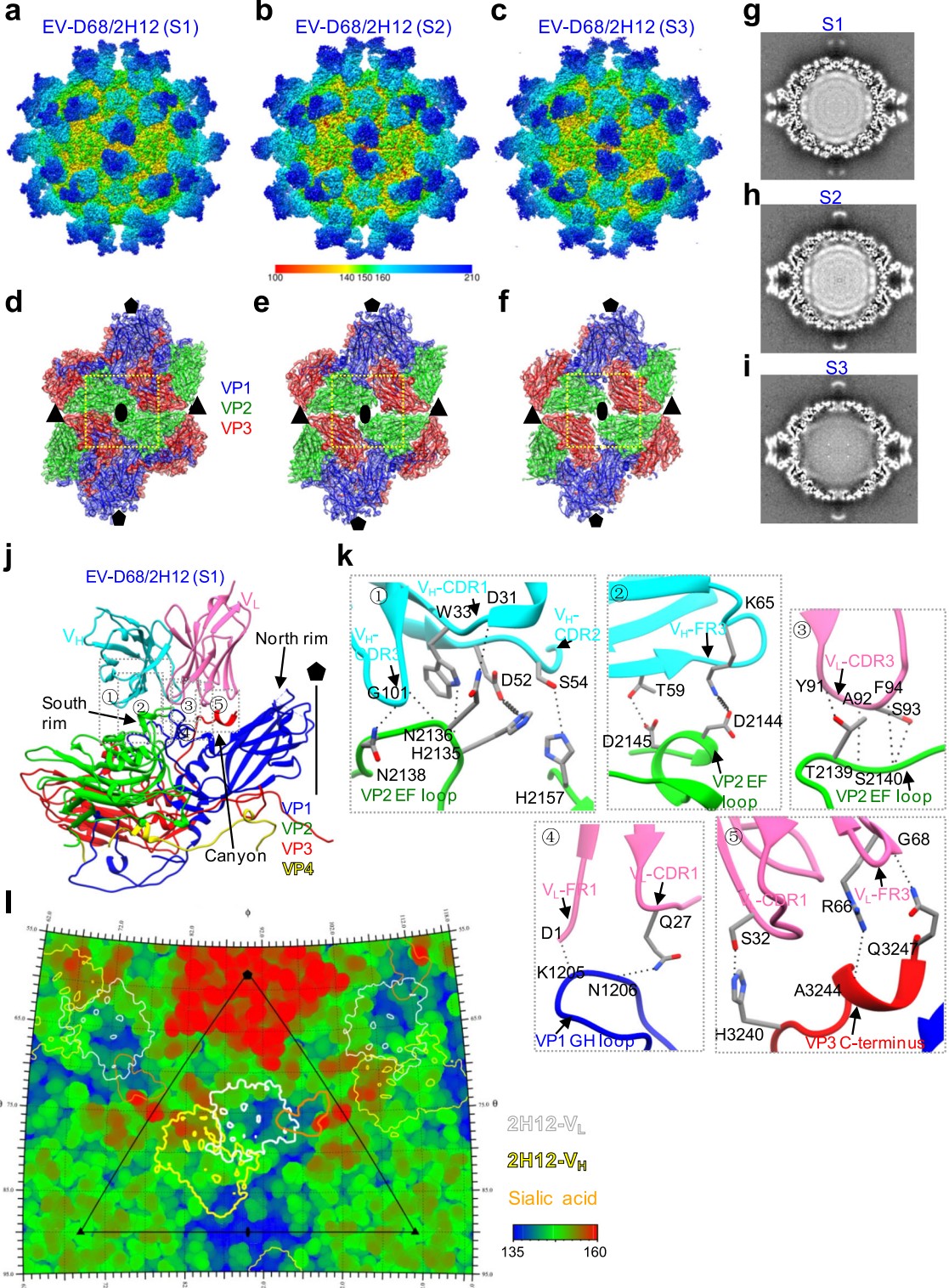

**Fig. 5 Cryo-EM structures of EV-D68 in complex with 2H12. a–c** Cryo-EM density maps of EV-D68/2H12, resolved in three different states (namely S1, S2, and S3), viewed along the two-fold axes. The color bar indicates the corresponding radius (unit in Å). **d–f** Density map with fitted model of the two-fold related protomers for the S1 (**d**), S2 (**e**), and S3 (**f**) states. Yellow dashed rectangles indicate the major differences among the three states. 2H12 Fab was removed for clarity. **g–i** Cryo-EM map central section of S1 (**g**), S2 (**h**), and S3 (**i**) states. **j** Binding Interface between EV-D68 protomer and 2H12 Fab in the S1 state. 2H12-V$_H$, 2H12-V$_L$, VP1, VP2, VP3, and VP4 are colored in cyan, hot pink, blue, green, red, and yellow, respectively. **k** Zoom-in view of the five dotted boxed regions in panel (**j**), showing interaction between VP2 EF loop, VP1 GH loop, and VP3 C-terminus of EV-D68 and the CDR and FR regions of 2H12 Fab. Black dashed lines indicate possible hydrogen bonds, and springs indicate both hydrogen bonds and salt bridges. **l** Roadmap shows footprint of 2H12 Fab (state S1) on the EV-D68 virion surface. The V$_L$ and V$_H$ are indicated by white and yellow contour lines, respectively. Footprint of the sialic acid receptor (generated using the PDB model 5BNO) is colored orange. The color bar indicates the corresponding radius (unit in Å).

Specifically, our S2 and S3 maps exhibit open channels at the two-fold axis (Fig. 5e, f), a characteristic feature of expanded enteroviral particles; whereas it appears closed in S1 (Fig. 5d) and in the native EV-D68 virion structure (PDB: 6CSG)[18]. Noteworthy, enterovirus genomes have been suggested to be released through the two-fold axis channel[30,31]. Moreover, the capsids in both S2 and S3 (163 Å and 162 Å in radius, respectively) are expanded relative to S1 and EV-D68/8F12 (both at 157 Å in radius) (Supplementary Fig. 8a). Interestingly, compared with S1 and S2, S3 shows different internal genomic RNA organization and significantly weaker RNA genome density (Fig. 5g–i).

Further structural comparison between the S2 and S3 states, by superimposition of their protomer structures together (Supplementary Fig. 8b-e), revealed the overall Cα RMSD value between the two structures to be 1.47 Å, indicating conformational variations between the two states. Specifically, the N-terminus of VP1 is externalized and the first 40 residues are missing in S3, while most of them could be resolved lying underneath the VP3 in S2; the VP1 C-terminal loop (residues K270 to T289) is largely missing in S3, while captured in S2 (Supplementary Fig. 8c). Noteworthy, VP4 is completely missing in S3, while part of VP4 is present in S2 structure (Supplementary Fig. 8b, 8d). Moreover, residues P43 to T54 from VP2 AB loop are also missing in S3 but captured in S2 (Supplementary Fig. 8d); residues P168 to T174 within VP3 GH loop display significantly different conformations in the two states (Supplementary Fig. 8e). Collectively, S2 and S3 are indeed in different states in capsid conformation.

In addition, we compared our structures with the available cryo-EM structures of the native mature virion (PDB: 6CSG), expanded 1 (E1, PDB: 6CS3), and A-particle (PDB: 6CS6) of the 18947 strain[18] (Supplementary Table 3). The overall Cα RMSD between the capsid protomer of the 18947 virion and that of our S1 state was calculated to be as small as 0.86 Å, indicating that the viral particle in S1 adopts the native conformation. A better fit in capsid protomer structure was observed between our S2 state and the 18947 E1 particle (RMSD = 1.21 Å) than between S2 and the 18947 native virion (RMSD: 1.99 Å) or between S2 and A-particle (RMSD: 3.52 Å), suggesting that S2 is more close to the E1 conformation instead of resembling the A-particle. Our S3 state fits better with the 18947 A-particle in the capsid protomer (RMSD: 1.61 Å) than with the other two structures (mature virion and E1 particle), but with significantly weaker RNA genome density (Fig. 5i) than typical A-particle (with genome RNA). It is worth noting that the N-terminus of VP1 and VP4 are located inside the capsid in S2 state and E1 particle, but the VP1 N-terminus is externalized and VP4 is lost in S3 state and A-particle[18] (Supplementary Fig. 8b-e). Taken together, these data indicated (1) the viral particle in S1 adopts the native configuration and remains intact; (2) the viral particle in S2 resembles the E1 conformation; (3) the viral particle in S3 represents a previously undescribed uncoating intermediate state, which likely occurs in the transition from the 135S A-particle (with genome RNA) to the 80S emptied particle (without genomic RNA). In summary, our structural analyses revealed that binding of 2H12 Fab can trigger EV-D68 virus uncoating.

Analyses of the interactions between EV-D68 virion and 2H12 in S1 state revealed that each 2H12 Fab binds to a single protomer (Fig. 5j); compared with 8F12 Fab, 2H12 Fab shows similar position and orientation on the virion surface (Figs. 4d, 5j). Specifically, 2H12 Fab contacts the south wall of the canyon, and its $V_L$ is tilted to the five-fold axis of EV-D68 virion, spatially covering the canyon and thus blocking the binding of the sialic acid receptor (Fig. 5j, l). Furthermore, the 2H12 epitope includes residues from the VP1 GH loop, VP2 EF loop, and VP3 C-terminus (Fig. 5k, Supplementary Table 5), among which five

residues are also involved in interactions between EV-D68 virion and 8F12. To be specific, the VP2 EF loop interacts with the CDR1, CDR2, CDR3, and FR3 of 2H12-$V_H$, as well as CDR3 of 2H12-$V_L$ (Fig. 5k①-③); the VP1 GH loop contacts with FR1 and CDR1 of 2H12-$V_L$ (Fig. 5k④); the VP3 C-terminus interacts with CDR1 and FR3 of 2H12-$V_L$ (Fig. 5k⑤). Putting together, EV-D68/2H12 interaction interface covers ~1155.9 Å$^2$ of the surface area, and the 2H12 heavy chain and light chain contribute 44.2% and 55.8% of the interaction interface, respectively (Supplementary Table 6).

**Comparison of the binding epitopes and neutralization mechanisms of 2H12 and 8F12.** For the EV-D68/2H12 (state S1) and EV-D68/8F12 immune complexes, their interaction interfaces are in similar sizes, but the interaction area between 2H12-$V_H$ and EV-D68 (511.1 Å$^2$) is significantly larger than that between 8F12-$V_H$ and EV-D68 (343.4 Å$^2$) (Supplementary Table 6). Superposition of the structures of EV-D68/2H12 (S1) and EV-D68/8F12 showed that the two immune complexes have broadly similar overall structures, but there are several apparent local structural differences between them (Fig. 6a). In particular, despite 2H12-$V_L$ and 8F12-$V_L$ appear overall similar in conformation (Fig. 6b), the 2H12-$V_H$ and 8F12-$V_H$ are quite different especially in the three CDR regions with the CDR1–3 of 2H12-$V_H$ appearing closer to the viral surface than the CDRs of 8F12-$V_H$ (Fig. 6c), leading to a larger interaction area between 2H12-$V_H$ and the virus.

We also superimposed the models of EV-D68/2H12 (S1), EV-D68/8F12, and native EV-D68 virion (PDB: 6CSG)[18] to analyze the structural variations in capsid proteins. Overall, the capsid proteins of EV-D68/8F12 and EV-D68/2H12 (S1) resemble closely those of native virion (Fig. 6d), still, several notable local conformational variations were observed for EV-D68/2H12 (S1). Upon 2H12 binding, one can observe a large shift in residues N206 to G218 region within the VP1 GH loop of EV-D68 which lies in the floor and south wall of the canyon (Fig. 6d). This 2H12 binding-induced conformational change of the VP1 GH loop might further lead to greater changes in the overall structure of EV-D68 virion, e.g., the expansion of the capsid structure and opening of the two-fold channel in S2 and S3 states.

In general, the binding footprints of 2H12 and 8F12 on the virion surface are very similar except that 8F12 can also interact with the VP1 BC loop (Figs. 4, 5), suggesting that the two antibodies may target the same antigenic site. To verify this structure-based prediction, we performed a BLI competitive binding assay, in which immobilized EV-D68 virion was saturated with the first antibody 2H12, and then allowed to bind the second antibody 8F12 in the presence of 2H12. As shown in Fig. 6e, the initial binding of 2H12 to EV-D68 virions greatly blocked subsequent 8F12 binding. Similarly, when MAb 8F12 was used as the first antibody, the second antibody 2H12 produced very low BLI signal (Fig. 6f). These data suggest that 2H12 and 8F12 compete with each other for binding to EV-D68 virion. Collectively, our structural and biochemical data demonstrate that the binding epitopes of 2H12 and 8F12 are similar or largely overlapping (Fig. 6g).

Both 2H12- and 8F12-binding footprints overlap with the sialic acid receptor binding site (Figs. 4d, f and 5j, l) and both MAbs can inhibit the binding of EV-D68 to sialic acid receptor in HI assays (Fig. 3), indicating that blockade of EV-D68 binding to sialic acid receptors via steric hindrance is the main neutralization mechanism shared by MAbs 2H12 and 8F12 (Fig. 6g). In addition, our structural data showed that 2H12 binding could destroy viral particles to some extent (Supplementary Fig. 6a) and trigger premature virus uncoating (Fig. 5g–i), likely leading to

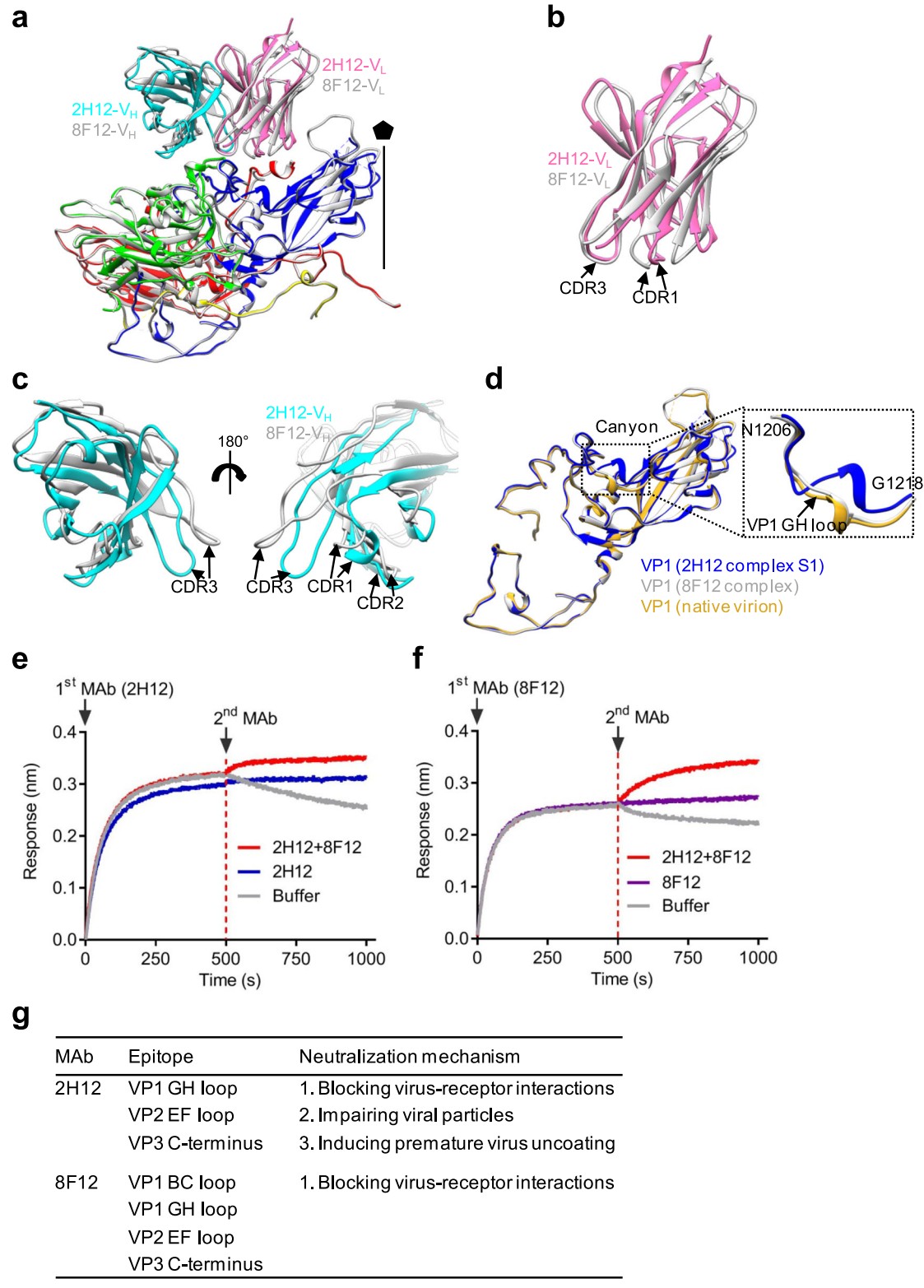

**Fig. 6 Structural comparison of the two immune complexes. a** Superposition of EV-D68/2H12 (state S1; color) and EV-D68/8F12 (gray) protomer structures. **b** Enlarged view of superposition of 2H12-V$_L$ (hot pink) and 8F12-V$_L$ (gray) structures. **c** Enlarged view of superposition of 2H12-V$_H$ (cyan) and 8F12-V$_H$ (gray) structures. **d** Superposition of the VP1 structures from EV-D68/2H12 (state S1; blue), EV-D68/8F12 (gray), and unbound native EV-D68 virion (PDB: 6CSG; goldenrod). The boxed area is enlarged to better illustrate the large movement of VP1 GH loop in EV-D68/2H12 (state S1). **e, f** Competitive binding assay using BLI. Immobilized EV-D68 virion (18947 strain) was saturated with the first antibody 2H12 (**e**) or 8F12 (**f**) and then allowed to react with the buffer, the antibody mixture (2H12 + 8F12), 2H12 alone (control) (**e**) or 8F12 alone (control) (**f**). **g** Summary of the epitopes and neutralization mechanisms of the 2H12 and 8F12 MAbs.

impaired virus infectivity. Therefore, MAb 2H12 may adopt two additional mechanisms to achieve neutralization, including impairing virion integrity and inducing premature virion uncoating (Fig. 6g).

**Human-mouse chimeric 2H12 and 8F12 antibodies are therapeutically efficacious.** Having determined the efficacy and working mechanisms of 2H12 and 8F12 antibodies, we then asked whether these two murine MAbs could be chimerized with human IgG constant regions. Human-mouse chimeric MAbs were created by separately linking the variable domains of murine 2H12 and 8F12 to the constant domains of human IgG1 heavy chain and human kappa light chain (Fig. 7a). The engineered human-mouse chimeric antibodies were successfully produced in HEK 293F cells (Fig. 7b). Chimeric MAbs 2H12 (c2H12) and 8F12 (c8F12) reacted with anti-human IgG (constant domains) antibody but not anti-mouse IgG (constant domains) antibody in western blot assays (Fig. 7b), confirming their chimeric nature. The purified c2H12 and c8F12 were >94% purity as determined by size exclusion chromatography (Supplementary Figure 9). Both c2H12 and c8F12 had high affinities to the 18947 virion with KD values below 14 nM (Fig. 7c, d, f). The results from neutralization assays showed that both c2H12 and c8F12 potently neutralized the EV-D68 strain 18947 with IC50s being less than 180 ng/mL (Fig. 7e, f). Taken together, these data indicate that the chimeric antibodies c2H12 and c8F12 retain high binding affinities and neutralizing activities.

The therapeutic potential of the chimeric MAbs was also assessed in the EV-D68 infection mouse model. Groups of 1-day-old ICR mice were inoculated with the 18947 strain and 1 day later given single injections of PBS, c2H12 (10 μg/g), or c8F12 (10 μg/g), respectively. Survival and clinical score were then monitored daily. As shown in Fig. 7g, h, all mice in the control (PBS) group died by 12 dpi; by contrast, 92.9% of the c2H12-treated mice and all of the c8F12-treated mice survived. These results indicate that both c2H12 and c8F12 retain excellent therapeutic efficacy.

## Discussion
EV-D68 infection can lead to severe respiratory illness and AFM[6,8]. Recent clinical surveys show that the prevalence of EV-D68 is increasing at a global scale[6,8], underscoring the need for effective therapeutic agents. Because multiple clades of EV-D68 currently exist and co-circulate[8] and a prodromal illness (such as febrile, respiratory, and/or gastrointestinal) preceded the onset of limb weakness in most AFM patients by a median of 5 days[32,33], a successful anti-EV-D68 therapeutic drug will need to be broadly effective against multiple EV-D68 stains/clades and remain efficacious even when administered at a delayed time point (eg. 3 days) in the course of infection. In the present study, we demonstrate that delayed treatment (3 dpi) with a two-MAb (2H12/8F12) cocktail robustly provides cross-clade protection in a mouse model of EV-D68 infection. This two-MAb cocktail meets the above mentioned requirements for an ideal anti-EV-D68 therapeutic agent and is therefore a potential drug candidate for further preclinical and clinical development.

Intriguingly, we found that 2H12 and 8F12 displayed distinct cross-neutralization profiles despite they were generated using the same immunogen (Fig. 1a). The mechanism underlying this interesting observation remains to be elucidated. Nonetheless, we discovered that the combination of 8F12 and 2H12 was able to potently neutralize multiple EV-D68 strains/clades, including strains 18947 (clade B), 18953 (clade D), and Fermon (prototype) (Fig. 1a). It is likely that 2H12 and 8F12 act independently and exert preferential neutralization on different virus strains, as the 2H12/8F12 combination was no more effective than the most

effective individual MAb for a given strain (eg. 2H12 for strain 18953 and 8F12 for strain Fermon) (Fig. 1a). However, the two MAbs in the cocktail nicely complement each other to achieve broad neutralization against diverse virus strains both in vitro and in vivo (Figs. 1a and 2), thus highlighting the advantage of the 2H12/8F12 antibody cocktail.

Here we demonstrated that a MAb-based therapy is effective in a mouse model of EV-D68 infection even when treatment is initiated at 3 days post-infection. The mouse model we used, in which the virus was inoculated via the intraperitoneal (i.p.) route, was established in our previous study[28]. This model produces not only robust infection as indicated by severe clinical manifestations, such as paralysis, and high viral loads in multiple tissues/organs including brain, lung and spinal cord, but also significant mortality[28]. Compared with a recently reported mouse model in which mice inoculated intranasally with EV-D68 developed only very transient viremia but no death[24], our i.p. infection model appears to be a more stringent tool for evaluation of the protective potential of antivirals and was therefore used in the present study. We should point out that, in this model, at 3 dpi EV-D68-inoculated mice had already displayed clinical signs such as reduced mobility and limb weakness (Fig. 2) and the virus had spread to multiple tissues/organs including brain and spinal cord[28], indicating a robust infection had established. Our results showed that a single dose of the 2H12/8F12 cocktail administered at 3 dpi protected almost all of the lethally challenged mice, demonstrating that the two-MAb cocktail is capable of reversing the disease caused by infection even when the virus has spread into the brain. We should emphasize that the 2H12/8F12 cocktail is effective on not only the clade B strain (18947) but also the clade D strain (18953). We did not evaluate MAb treatment at later time points such as 5 dpi, because some of the mice in the control group (without anti-EV-D68 MAb treatment) already died at 5 dpi (Fig. 2). Nonetheless, our work demonstrates that the 2H12/8F12 cocktail possesses remarkable therapeutic potency and breadth, warranting its further development as an anti-EV-D68 therapy.

Importantly, our high-resolution cryo-EM structures revealed a previously undescribed neutralization antigenic site on EV-D68 capsid. Unlike the previously reported anti-EV-D68 MAbs which epitopes reside around the five-fold or three-fold axes of viral capsid[19,24,25], both of our 2H12 and 8F12 MAbs bind to the south rim of the canyon around the two-fold axes (Figs. 4d, 5j). Accordingly, both 2H12- and 8F12-binding footprints overlap with the identified sialic acid binding site (Figs. 4f, 5l), creating strong steric hindrance on sialic acid receptor binding. This structural observation is consistent with the experimental results that both 2H12 and 8F12 efficiently inhibited EV-D68 attachment to susceptible cells and blocked the interaction between EV-D68 and sialic acid in the HI assays (Fig. 3). Collectively, these data demonstrate that blockade of EV-D68 binding to cellular sialic acid receptor via steric hindrance is a major neutralization mechanism shared by 2H12 and 8F12 MAbs.

Moreover, an interesting finding here is that 2H12 binding can trigger virus uncoating. MAb 2H12 was found to impair EV-D68 viral particles to some extent (Supplementary Fig. 6a), induce capsid expansion, and cause premature release of a portion of viral RNA from the capsid (Fig. 5, Supplementary Fig. 8), probably leading to decreased viral infectivity. Several neutralizing antibodies against picornaviruses, such as anti-enterovirus 71 (EV71) antibodies E18[34], A9, and D6[35], anti-human rhinovirus B14 antibody C5[36], and anti-EV-D68 antibody 15C5[19], have been reported to have the ability to induce viral uncoating. It is worth noting that all of these antibodies bind to regions surrounding the three-fold axes of viral capsid and across the interface between protomers. However, for our antibody 2H12, its binding site lies near the canyon and around the two-fold axes (Fig. 5), suggesting 2H12 may use a distinct mechanism to induce virus uncoating. It

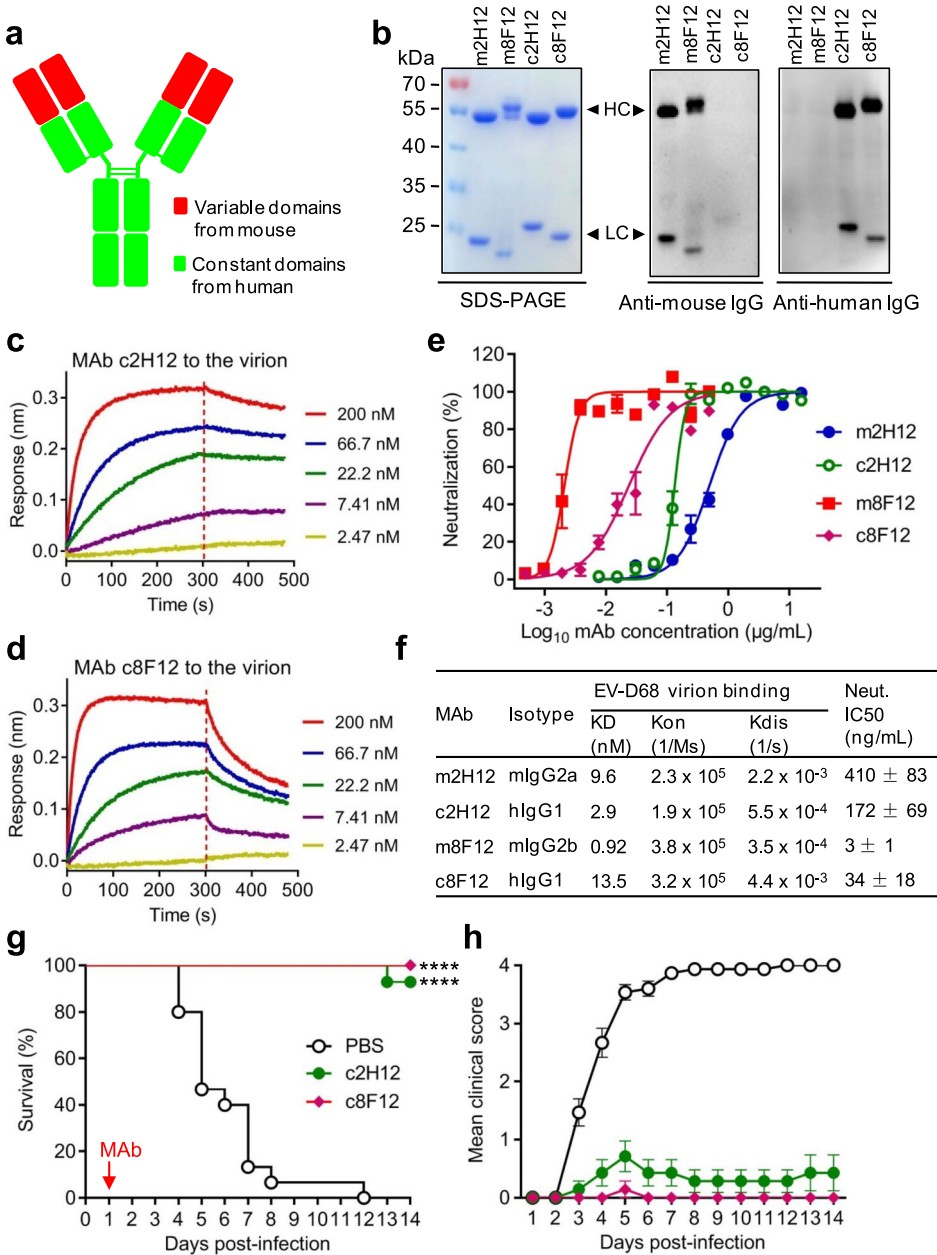

**Fig. 7 Binding affinities, neutralization activities and therapeutic efficacy of the chimeric MAbs. a** Schematic of chimeric MAb. Mouse variable domains are shown as red boxes, while human constant domains are shown as green boxes. **b** SDS-PAGE and western blotting analysis of the purified MAbs. Lane M, protein marker; m2H12, murine MAb 2H12; m8F12, murine MAb 8F12; c2H12, chimeric MAb 2H12; c8F12, chimeric MAb 8F12. Anti-human IgG or anti-mouse IgG secondary antibodies (HRP conjugate) used in this study recognize the constant domains of both heavy (HC) and light chains (LC). **c, d** Binding affinities of the chimeric MAbs c2H12 (**c**) and c8F12 (**d**) to EV-D68 18947 virion measured by BLI. MAb concentrations used were shown. **e** Comparison of neutralization activities of the chimeric (c2H12 and c8F12) and murine (m2H12 and m8F12) MAbs against EV-D68 strain 18947. The method was described in the legend of Fig. 1. Result shown is representative of three independent experiments. **f** Characteristics of chimeric MAbs. KD values for the interaction of the MAbs with EV-D68 18947 virion were determined by BLI as described in panels (**c, d**) and Fig. 1d, e. Half inhibitory concentration (IC50) of each antibody was calculated using GraphPad Prism software and results represent the mean ± SD of three independent experiments. Neut., neutralization. **g, h** Therapeutic efficacy of chimeric MAbs against EV-D68 infection in neonatal mice. One-day-old ICR mice ($n = 14$–15/group) were inoculated i.p. with strain 18947. The suckling mice were i.p. injected with PBS, 10 μg/g of c2H12, or 10 μg/g of c8F12 at 1 dpi. Red arrow indicates time of MAb administration. Clinical scores were graded described in Fig. 2 legend. Survival of mice in each antibody-treated group was compared to the PBS control group. Statistical significance was determined by Log-rank (Mantel-Cox) test and was indicated as follows: ****, $p < 0.0001$. In panel (**g**), $p$ value between the c2H12 or c8F12 group and the PBS control group is below 0.0001. Error bars represent SEM.

is intriguing that 2H12, but not 8F12, can induce virus uncoating, despite both MAbs bind the same region on EV-D68 capsid. Structural comparison showed that a large movement in residues N206 to G218 within the VP1 GH loop of EV-D68 occurs in the virion/2H12 complex (S1) but not in the virion/8F12 complex

(Fig. 6d). Such a large local movement, which is probably due to the formation of hydrogen bond between N206 [O] of VP1 and Q27 [NE2] of 2H12 light chain (CDR1) (Fig. 5k④), may prime subsequent greater changes in the overall structure of EV-D68 virion, such as capsid expansion, opening of the two-fold channel

and RNA release. It is worth mentioning that, in EV71, the VP1 GH loop, where EV71 uncoating receptor scavenger receptor class B member 2 (SCARB2) binds[37], has been suggested to act as an adapter-sensor for cellular receptor attachment and play an important role in viral uncoating[37,38]. Hence, it is likely that our antibody 2H12 may use a receptor mimic mechanism to initiate viral particle transformation and virus uncoating. Nonetheless, our results demonstrate that 2H12 can neutralize EV-D68 infection via multiple mechanisms, including blocking virus attachment to the cell surface through interfering with the virus/ sialic acid interaction, impairing viral particles, and inducing premature virus uncoating.

In summary, our study demonstrates proof-of-concept of a chimeric two-MAb cocktail for delayed treatment of diverse EV-D68 infections and elucidates its working mechanism and structural basis, paving the way for developing a MAb-based pan-EV-D68 therapy for human use.

## Methods

**Cells and viruses**. Human rhabdomyosarcoma (RD) cells were grown in DMEM (Gibco, USA) supplemented with 5% fetal bovine serum (FBS). Mouse myeloma cell line SP2/0 was cultured in RPMI 1640 medium (Life technologies, USA) supplemented with 10% FBS at 37 °C. EV-D68 prototype strain Fermon, and two clinical isolates US/MO/14-18947 (hereinafter referred to as 18947) and US/KY/14-18953 (hereinafter referred to as 18953) were described in Supplementary Table 1. All viruses were expanded in RD cells and titrated by the 50% tissue culture infectious dose (TCID$_{50}$) assay, according to the Reed and Muench method[39].

**Antigens and antibodies**. EV-D68 VLP was produced in insect cells in our previous study[26]. EV-D68 18947 viral particles were prepared as described previously[28]. Enterovirus 71 (EV71) VLP and coxsackievirus A16 (CVA16) VLP were separately generated in insect cells in our previous study[40].

MAb D5 is an antibody against EV71[41]. MAb 9B5 is an antibody against CVA16[42]. MAb 1C11 is an IgG2a antibody against zika virus (ZIKV) E protein[43] and MAb 1F4 is an IgG2b antibody against hepatitis C virus (HCV) E2 protein, serving as isotype controls in this study.

**Preparation of MAbs and Fabs**. The animal studies were approved by the Institutional Animal Care and Use Committee at the Institut Pasteur of Shanghai, and the project identification code was A2018040. All mice were purchased from Shanghai Laboratory Animal Center (SLAC, China).

Before immunization, purified EV-D68 VLP (5 μg/dose) was thoroughly mixed with aluminum hydroxide adjuvant (500 μg/dose; Invivogen, USA). Female BALB/c mice aged 6–8 weeks were injected intraperitoneally (i.p.) with aluminum-adsorbed VLP antigen at weeks 0, 2, and 4. The mice were then boosted by tail vein injection with 10 μg of VLP in PBS. Splenocytes were harvested 3 days after the boost and fused with SP2/0 myeloma cells using polyethylene glycol (PEG) 1450 (Sigma, USA). Fused cells were cultivated in hypoxanthine, aminopterin, and thymidine (HAT; Sigma) selective growth medium for 8 days. The following hybridoma supernatants were screened for the ability to neutralize EV-D68 infection by neutralization assay as described below. Positive hybridoma cells were cloned 2–4 times by limiting dilution to obtain monoclonal cell lines. Purified MAbs were prepared from ascitic fluids using protein G agarose resin 4FF (Yeasen, China). Hybridoma cells were preserved at −80 °C using CELLSAVING (New Cell & Molecular Biotech, Suzhou, China).

To prepare Fabs, purified anti-EV-D68 MAb 2H12 was digested with papain using a previously published protocol[44]. The resultant 2H12 Fab was purified by affinity chromatography using HiTrap™ Protein L column (GE Healthcare, USA). Recombinant His-tagged Fab fragment of anti-EV-D68 MAb 8F12 was produced in HEK (Human Embryonic Kidney) 293F suspension cells according to the protocol described in a previous study[45] and then purified by using Ni$^{2+}$-resins. The purified 8F12 Fab sample was adjusted to pH ~9.0 with 7.5% (w/v) NaHCO$_3$ before use.

**Neutralization assay**. Neutralizing activities of hybridoma supernatants and purified MAbs against EV-D68 were determined by micro-neutralization assay as described previously[27]. Neutralization concentration of the MAbs was defined as the lowest antibody concentration that could fully inhibit cytopathic effect (CPE). Immediately after the observation of CPE, cell viability was determined using CellTiter-Glo 2.0 assay kit (Promega, USA) according to the manufacturer's instructions. Percent neutralization was calculated as follows: 100 x (luminescence of the given sample−luminescence of the virus-only sample)/(luminescence of the cell-only sample−luminescence of the virus-only sample). Half inhibitory concentration (IC50) of each MAb was calculated using GraphPad Prism software by nonlinear regression. IC50 was defined as the antibody concentration required inhibiting virus infection by 50% compared to the virus-only sample.

**ELISA**. Isotypes of the MAbs were determined by ELISA using the SBA Clonotyping System-HRP kit (Southern Biotech, USA) according to manufacturer's instructions.

To determine binding specificity of the MAbs, microplates (Nunc, USA) were coated at 4 °C overnight with 50 ng/well of EV-D68 VLP, EV71 VLP[40], or CVA16 VLP[40] and then blocked with 5% milk in PBS-Tween20 (PBST). After washes with PBST, 50 μL/well of two-fold serially diluted purified MAbs was added to the wells, followed by 2 h incubation at 37 °C. After washing, horseradish peroxidase (HRP)-conjugated anti-mouse IgG (diluted 1:10,000; Sigma-Aldrich, USA) was added and incubated for 1 h at 37 °C. After color development, absorbance was monitored at 450 nm with a microplate reader.

**Determination and analysis of MAb sequences**. To identify antibody sequences, total RNA was isolated from hybridoma cells using TRIzol reagent (Invitrogen, USA). First-strand cDNA was then synthesized, purified and tailed using the 5′ RACE System (Invitrogen) following the manufacturer's protocols. Tailed cDNA is amplified by PCR with Ex Taq (Takara, Japan) and then cloned into pMD19-T vector (Takara) for sequencing, to obtain 5′-terminal sequences of both heavy and light chains of the MAbs.

Positions of complementarity determining regions (CDR) and the closest mouse IgG germline genes were identified using the IgBLAST tool[46].

**Bio-layer interferometry (BLI) assay**. Before BLI assay, purified EV-D68 18947 viral particles[28] were labeled with EZ-Link™ Sulfo-NHS-LC-LC-Biotin (Thermo Fisher Scientific, USA) following manufacturer's instructions and then purified using Zeba™ spin desalting column (Thermo Fisher Scientific) to remove excess non-reacted biotin. To determine binding affinity of the antibodies to EV-D68, BLI assay was performed in an Octet® RED96 System (Pall FortéBio, USA) according to manufacturer's instructions. Briefly, biotinylated EV-D68 18947 virion was immobilized onto streptavidin-coated biosensors (Pall FortéBio) until saturation. The antigen-bound biosensors were placed in wells containing a series of diluted MAb samples to allow antigen-antibody association and were then dipped into dissociation buffer (0.01 M PBS supplemented with 0.1% bovine serum albumin and 0.02% Tween 20). Equilibrium dissociation constants (KD) were calculated using Octet data analysis software version 11.0 (Pall FortéBio).

For competitive binding assay, the immobilized EV-D68 18947 virion were placed in wells containing 30 μg/mL of the first antibody for 500 s (association phase). The sensors were then immersed into wells containing dissociation buffer, 2H12 or 8F12 alone (30 μg/mL; control) or the antibody mixture (30 μg/mL of 8F12 plus 30 μg/mL of 2H12) for 500 s. The data were then analyzed using Octet data analysis software (Pall FortéBio).

**In vivo protection assays**. The animal studies were approved by the Institutional Animal Care and Use Committee at the Institut Pasteur of Shanghai and the project identification code was A2018040.

The prophylactic and therapeutic efficacy of anti-EV-D68 neutralizing MAbs was assessed in a mouse model of EV-D68 infection[28]. For the prophylactic experiment, groups of naive ICR mice (age <24h) were i.p. administrated with PBS, 10 μg/g of anti-EV-D68 MAbs (2H12 or 8F12), or 10 μg/g of isotype control MAbs (1C11 or 1F4), and 1 day later the mice were infected with $8.0 \times 10^4$ TCID$_{50}$ of strain 18947.

For the therapeutic experiments, groups of 1-day-old ICR mice were inoculated i.p. with $8.0 \times 10^4$ TCID$_{50}$ of strain 18947 or $1.0 \times 10^6$ TCID$_{50}$ of strain 18953, and 24 h or 72 h later the mice were i.p. injected with PBS, 10 μg/g of 2H12, 10 μg/g of 8F12, or a mixture of both 2H12 and 8F12 (10 μg/g of each MAb). After infection, all mice were observed daily for survival and clinical score for 2 weeks. Clinical scores were graded as follows: 0, healthy; 1, lethargy and reduced mobility; 2, limb weakness; 3, limb paralysis; 4, death.

**Pre- and post-attachment inhibition assays**. For pre-attachment inhibition assay, 1000 TCID$_{50}$ of strain 18947 was mixed with 1 μg of the MAbs, and the mixtures were added to prechilled RD cells in 24-well plates and kept at 4 °C for 1 h to allow virus attachment. The cells were then washed with ice-cold PBS, and fresh DMEM supplemented with 1% FBS was added and incubated at 33 °C. For post-attachment inhibition assay, 1000 TCID$_{50}$ of strain 18947 was added to cooled RD cells and incubated for 1 h at 4 °C for virus adsorption. After washing twice to remove unbound virus, fresh culture media was added and incubated at 33 °C for various time periods (0, 0.5, 1, 2, or 5 h) to allow virus entry followed by treatment with 1 μg of the MAbs. For both assays, the cells and culture supernatants were collected together 8 h after infection and lysed in TRIzol reagent (Invitrogen) for RNA extraction. Total RNA was reverse transcribed using PrimeScript RT reagent Kit (Takara, Japan), and real-time PCR was performed using SYBR Premix Ex Taq kit (Takara) according to the manufacturer's protocols. Primers for strain 18947 were as follows: forward primer, 5′-CGAGAGCATCATCAAAACAGCGACC-3′; reverse primer, 5′-CACTGTGCGAGTTTGTATGGCTTCT-3′. β-actin primer sequences were as follows: forward primer, 5′-GGACTTCGAGCAAGAGATGG-3′; reverse primer, 5′-AGCACTGTGTTGGCGTACAG-3′. Data analysis was performed using the $2^{-\Delta\Delta Ct}$ method[47] with β-actin as the internal control.

**Inhibition of virus attachment by the antibodies**. In total, $1.0 \times 10^7$ TCID$_{50}$ of EV-D68 strains 18947 or 18953 was incubated with increasing amounts (0.01 μg, 0.1 μg, or 1 μg) of the MAbs at 37 °C for 1 h. The mixtures were then cooled to 4 °C and added to prechilled RD cell monolayers grown in 24-well plates followed by incubation at 4 °C for 1 h to permit virus attachment. The cells were washed three times with ice-cold PBS and harvested with TRIzol reagent (Invitrogen) for RNA extraction. RNA was subjected to real-time reverse transcription PCR (RT-PCR) as described above. Primers for 18947 and β-actin were described above. Primers for 18953 strain were as follows: forward primer, 5′-GGAAGCCATACAAACTCG-3′; reverse primer, 5′-TTCGTGCTTCAGATGAGGTG-3′. Data analysis was performed using the $2^{-\Delta\Delta Ct}$ method[47] with β-actin as an endogenous control.

**Hemagglutination inhibition (HI) assay**. The hemagglutination unit (HAU) of purified EV-D68 18947 virion was determined by hemagglutination assay. Briefly, EV-D68 18947 virion was two-fold serially diluted with PBS in a 96-well V-bottom microtiter plate. 50 μL of 1% chicken red blood cell (RBC) suspension was added to each well and incubated for 0.5 h at room temperature. HAU was read as the highest dilution of virion exhibiting complete hemagglutination

In HI assay, 25 μL/well of two-fold serially diluted purified MAbs was mixed with 25 μL/well of 8 HAU of EV-D68 18947 virion, followed by 1 h incubation at RT. 50 μL of 1% RBC suspension was added to each well and incubated for 0.5 h at room temperature. HI concentration of the MAbs was defined as the lowest antibody concentration that could fully inhibit virus-induced hemagglutination.

**Cryo-EM imaging**. To prepare immune complexes, EV-D68 18947 virion, and 2H12 or 8F12 Fab were incubated at a molar ratio of 1:120 for 20 min at room temperature. 3 μL of EV-D68/2H12 or EV-D68/8F12 complex was placed onto a plasma-cleaned holey carbon grid (R1.2/1.3, 200 mesh; Quantifoil Micro Tools) or a continuous ultrathin carbon film covered lacey carbon grid (400 mesh; Ted Pella), respectively. The grids were blotted and plunged into liquid nitrogen-cooled liquid ethane with a Mark IV Vitrobot (Thermo Fisher Scientific). Cryo-EM movies of the samples were collected on a Titan Krios transmission electron microscope (Thermo Fisher Scientific) operated at an accelerating voltage of 300 kV. The movies were recorded using a K2 Summit direct electron detector (Gatan) in super-resolution counting mode (yielding a pixel size of 1.318 Å after two times binning), in an automatic manner by SerialEM[48]. Each movie was dose-fractioned into 38 frames. The total electron dose was set to ~38 e$^-$/Å$^2$. The exposure time for each frame was set to 0.2 s. Defocus values ranged from −0.4 to −1.8 μm (Supplementary Table 2).

**Cryo-EM single particle 3D reconstruction**. Single-particle analysis was mainly executed using RELION 3.0[49]. Movie frames were aligned and summed using MotionCor2 program[50]. After CTF parameter determination using CTFFIND4[51] and Gctf[52], particle auto-picking, manual particle checking, and several rounds of reference-free 2D classification, 35,004 EV-D68/8F12 particles and 8844 EV-D68/2H12 particles remained for further processing.

A coxsackievirus A10 reconstruction[53] low pass filtered to 40 Å resolution was used as initial model for the first-round reconstruction of EV-D68/8F12 complex, and the resulting map was used as a model for further reconstructions. Only one conformational state was found in the EV-D68/8F12 dataset through 3D classification (Supplementary Fig. 4). After CTF refinement and Bayesian polishing, the particles went through another round of 2D classification with 30,540 cleaned-up particles remained. After another round of CTF refinement, 3D auto-refine, and Ewald sphere correction[54], we obtained a 2.89 Å resolution map of EV-D68/8F12 complex. The overall resolution was determined based on the gold-standard criterion using an FSC of 0.143[55].

Similar processing tactics were applied to EV-D68/2H12 dataset (Supplementary Fig. 6). After 2D classification, 8,844 particles remained. After CTF refinement and Bayesian polishing, these particles were reconstructed to a 3.25 Å resolution map. Further 3D classification of this dataset into three classes revealed two distinct major conformations with better structural features (class 1 and 3, Supplementary Fig. 6a). Class 1 map depicts the full native virion feature and no other conformation could be detected after further 3D classification, which was thus termed S1 state. Meanwhile, class 3 map exhibits open channels at the two-fold axis, a characteristic feature of expanded enteroviral particles. We then performed further 3D classification on this dataset and obtained two distinct conformational states (termed S2 and S3, Supplementary Fig. 6a). Compared with S2, S3 shows different internal genomic RNA organization and significantly weaker RNA genome density. Although the overall capsid structures of S2 and S3 resemble each other, there are still some differences, for example, density for VP4 is totally missing in S3, which is not the case in S2. After CTF refinement, Bayesian polishing, and Ewald sphere correction, the final resolutions of the S1, S2, and S3 maps were 3.09 Å, 3.60 Å, and 3.60 Å, respectively, based on the gold-standard criterion using an FSC of 0.143[55].

**Model building**. Previously published cryo-EM structures of EV-D68 strain 18947[18] were used as initial models. Especially, full native virion structure (PDB: 6CSG) was used for building the models of EV-D68/8F12 and EV-D68/2H12 (S1),

E1 particle structure (PDB: 6CS3) for EV-D68/2H12 (S2), and E1 particle structure (PDB: 6CS3) together with A-particle structure (PDB: 6CS6) for EV-D68/2H12 (S3). Meanwhile, homology models of 8F12 and 2H12 Fab were built through the SWISS-MODEL webserver[56]. Note that only the better resolved variable domains were kept, while the constant regions were removed from the models. Phenix real-space refinement[57] was used to refine model against corresponding map, followed by local manual adjustment by COOT[58]. The final atomic models were validated by phenix.molprobity[59]. The validation statistics of the atomic models are summarized in Supplementary Table 2. Figures were generated using UCSF Chimera[60]. Fab-virion interaction analysis including hydrogen bond, salt bridge prediction, and buried surface area calculation were carried out through PISA server (https://www.ebi.ac.uk/pdbe/prot_int/pistart.html)[61]. Road-maps were generated by RIVEM (Radial Interpretation of Viral Electron density Maps)[62]. Footprint for sialic acid is generated using the EV-D68-6′SLN model (PDB: 5BNO)[21].

**Preparation and characterization of chimeric MAbs**. To generate chimeric MAbs, heavy and light chain variable region genes of anti-EV-D68 MAbs were separately cloned into a modified pcDNA3.4 vector that contains interleukin-10 (IL-10) signal sequence and constant region genes of human immunoglobulin (gamma 1, kappa) using ClonExpress II One Step Cloning Kit (Vazyme, China). Primer information is listed in Supplementary Table 7. The resulting light and heavy chain expression plasmids were co-transfected into HEK 293F suspension cells, yielding chimeric antibodies. The recombinant chimeric MAbs were purified and analyzed using size exclusion chromatography to determine other contaminates, aggregates, and purity. The chimeric MAbs were subjected to BLI and neutralization assay as described above. In vivo therapeutic efficacy of the chimeric MAbs against strain 18947 was tested in the mouse model as described above.

**Western blotting**. Western blotting of chimeric MAbs was carried out as described previously[63], except that HRP-conjugated goat anti-mouse IgG (diluted 1:10,000; Sigma, USA) and HRP-conjugated rabbit anti-human IgG (diluted 1:10,000; Abcam, USA) were used for detection. Note that anti-human IgG or anti-mouse IgG secondary antibodies (HRP conjugate) used in this study recognize the constant domains of both heavy and light chains.

**Statistical analysis**. All statistical analyses were performed using GraphPad Prism version 8.

**Reporting summary**. Further information on research design is available in the Nature Research Reporting Summary linked to this article.

## Data availability

The authors declare that all relevant data are available from the corresponding authors upon reasonable request. Cryo-EM map determined in the EV-D68/8F12 dataset has been deposited at the Electron Microscopy Data Bank with accession code of EMD-31056, and associated atomic model has been deposited in the Protein Data Bank with accession code of 7EC5. Cryo-EM maps (S1, S2, and S3) determined in the EV-D68/2H12 dataset have been deposited at the Electron Microscopy Data Bank with accession codes of EMD-31055, EMD-31054, and EMD-31060, respectively, and related models have been deposited in the Protein Data Bank under accession codes of 7EBZ, 7EBR, and 7ECY, respectively. The sequences of 2H12-VH, 2H12-VL, 8F12-VH, and 8F12-VL have been deposited in GenBank with the accession codes MW627209, MW627210, MW627211, and MW627212, respectively. Source data are provided with this paper.

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

## Acknowledgements

We are grateful to the staff of the NCPSS Electron Microscopy facility and Database and Computing facility for instrument support and technical assistance. Z.H. was supported by grants from the Strategic Priority Research Program of the Chinese Academy of Sciences (XDB29040000), the International Cooperation Program of the Chinese Academy of Sciences (153831KYSB20170043), the National Natural Science Foundation of China (31872747), the TOTAL Foundation, the K.C.Wong Education Foundation, and the Youth Innovation Promotion Association of CAS (2016249). H.K.W was supported by the grants from National Key R&D Program of China (2016YFA0502202), Strategic Priority Research Program of the Chinese Academy of Sciences (XDB29030103). Y.C. was supported by the grants from the Strategic Priority Research Program of CAS (XDB37040103), the National Basic Research Program of China (2017YFA0503503), the National Natural Science Foundation of China (31670754 and 31872714), Shanghai Academic Research Leader (20XD1404200), CAS Facility-based Open Research Program and the CAS-Shanghai Science Research Center (CAS-SSRC-YH-2015-01, DSS-WXJZ-2018-0002). C.Z. was supported by China Postdoctoral Science Foundation (2018M633026), the National Natural Science Foundation of China (31800777), the Youth Innovation Promotion Association of the Chinese Academy of Sciences (CAS), and a postdoc fellowship from the Guangzhou Women and Children's Medical Center. We thank BIOINTRON company (China) for its help in recombinant antibody expression. We are grateful to Dr. Chuan Xiao (University of Texas at EI Paso) for the instruction of RIVEM, and Dr. Zhenguo Chen (Fudan University) for providing the continuous carbon covered Lacy Carbon grids.

## Author contributions

Z.H., Y.C., C.Z., C.X., W.L.D., and S.T.G. designed experiments. C.Z., C.X., W.L.D., Y.F.W., Z.L., X.Y.Z., and X.S.W. performed research. Z.H., Y.C., C.Z., C.X., and W.L.D. analyzed data. H.K.W advised the research; Z.H., Y.C., C.Z., and C.X. finalized the manuscript.

## Competing interests

C.Z., W.L.D., and Z.H. are listed as inventors on pending patent applications for MAbs 2H12 and 8F12. The other authors declare no competing interests.
