## [Peer Review File · Nature Communications]

REVIEWER COMMENTS

Reviewer #1 (Remarks to the Author):

In this manuscript Zhang, et al detail a study of two murine monoclonal antibodies (mAbs) isolated by the authors after immunization of mice with enterovirus D68 (EV-D68) virus-like particles. These two mAbs have unique neutralization profiles across different clades of EV-D68 despite binding to nearly identical epitopes as identified by the authors using cryo-EM. Further, they performed mechanistic studies to identify how the antibodies neutralize EV-D68, by blocking interactions with sialic acid and inhibiting attachment to cells in vitro. The cryo EM structures also indicate that one of the mAbs, 2H12, may additionally neutralize by capturing a novel structural intermediate of the virus that can allow escape of the genomic material. Finally, they show that both native murine and chimeric human forms of the mAbs can protect mice from EV-D68 challenge when given up to 3 or 1 days after infection, respectively.

The epitopes identified are unique from the only other structurally identified EV-D68 mAb epitopes in the literature: two murine mAbs in reference 19 and two human mAbs in reference 25. The macaque mAb epitope in reference 24 was not confirmed with structural methods. This represents an advance in the understanding of how antibodies can interact with EV-D68, a medically significant virus responsible for causing outbreaks of a severe paralyzing illness, acute flaccid myelitis. The difference in neutralization mechanisms despite nearly overlapping epitopes is also of general interest in the understanding of antibody-virus interactions.

The work in this paper is quite thorough and generally well arranged and easy to follow. There are a number of statements of primacy, though, that are unfounded or unnecessary. This is especially noted with the use of "elite" in line 408 and throughout lines 72-83. For example, the study in reference 25 does in fact test the therapeutic window of giving mAbs at different time points after infection in mice, which is counter to the claims in lines 78-79. Based on this, the use of "unprecedented" in line 455 is not applicable.

It is also curious that the discussion in lines 410-431 compares the mAbs in this study to prior referenced murine and macaque mAbs but not the human mAbs, which have equivalent to stronger neutralization potency against the same EV-D68 isolates. In any case, it is difficult to quantitatively compare neutralization potencies across different studies because they are measured using different techniques in different laboratories. The studies in this paper are of sufficient interest and quality that the comparisons are not necessary to bolster the significance.

The statement in lines 220-222 could be reworded to improve accuracy, as the mAb cocktail was never superior to a single mAb in any of the in vivo experiments. This statement and the use of the word "synergy" in line 159 seem to imply that the antibody cocktail of 2H12 and 8F12 is superior in efficacy to either mAb individually. That would demonstrate synergy. What the authors show is simply that each antibody individually is more effective at neutralizing or protecting against some EV-D68 strains than others. Therefore, giving a cocktail of mAbs provides redundancy, but there is no synergy. The combination of mAbs is no more effective than the most effective individual mAb in the cocktail.

The authors use the term "humanized" (first used in line 374) in a manner inconsistent with the antibody field. The antibodies used in this study are mouse-human chimeras, or "chimeric" antibodies. The term humanized more specifically implies that only the CDR loops of the murine mAb were used in place of the CDRs of a human mAb backbone. The authors swapped out the entire variable regions, though, which is not truly "humanization." This has important clinical implications because truly humanized mAbs are less immunogenic to humans than the mouse-human chimeras described in this manuscript. Variations of the word humanized should be eliminated from the manuscript and replaced with variations of the word chimeric.

-Description of chimeric antibodies: <https://absoluteantibody.com/our-technology/formats-we-have-made/chimeric-antibodies/>

-Description of how palivizumab was humanized, swapping out only the CDRs: S Johnson et al, J Infect Dis, 1998; PMID: 9359721 DOI: 10.1086/514115

The authors should further clarify that the specific antibodies they use for the western blot in Fig. 7b, discussed in lines 380-382, are Fc-specific. This is not mentioned in the methods or in the main text. One would otherwise expect a polyclonal anti-mouse IgG antibody to react to both the native murine mAb and the human-chimeric mAb.

The authors frequently refer to reference 32 (lines 401 and 432) in describing a 1-3 day incubation period for EV-D68 in humans. This reference is a small study that specifically refers to respiratory illness. The therapeutic mouse models described in this manuscript focus on a neurologic phenotype (paralysis) related to EV-D68 infection. Much larger studies have described median 5 days of illness preceding limb weakness in AFM, the typical human neurologic phenotype of EV-D68 infection. This longer period of incubation to AFM should be acknowledged in the manuscript.

-Study showing 5 day incubation: J Lopez et al, MMWR 2019, PMID: 31295232 DOI: 10.15585/mmwr.mm6827e1

There are a few graphs in the paper that appear to be lacking error bars, specifically the ELISA in Fig. s2 and the clinical scores in Figs. 2, S3B, and 7H. Regarding the clinical scores, it is impossible to understand the heterogeneity in clinical disease in the mice without them. Alternatively, the authors could create a supplemental table listing the clinical scores for each mouse to give an idea of this heterogeneity.

Reviewer #2 (Remarks to the Author):

This is an important work to characterize EV-D68 neutralizing Mabs, 2H12 and 8F12. Their efficacy as a cocktail for treatment of infections is explored by multiple assays including a structural characterization. The largest concern is a lack of careful quantification to accompany the structural analysis. There are a few other minor concerns, necessary clarifications, and areas where more information is needed.

Line 93 and elsewhere, 2.9 is not atomic resolution and should be described as high resolution or near atomic resolution.

Line 142 Bio-layer interferometry (BLI) is not defined or cited when first used in the Results.

Manuscript would greatly benefit from a table listing the different strains and each source and abbreviation used throughout.

There is not enough description for the cryo EM section. Description of the unchanged virus capsid should include quantification. What is the RMSD of the capsid superimposed with virus versus A-particle structures? Such a comparison should be the basis of a conclusion that the FAB does not induce change, not a visual inspection and speculation that the two fold is closed or open.

Line 316, in all reports to data, genome release at the two fold is suggested. This is a model not proven yet, but merely suggested by numerous A-particle structures.

Include a central section for 8F12 complex map as well so that the quality of the density can be seen.

Building the variable domains of the antibody models should be described prior to reporting on the

orientation of the light and heavy chains relative to the capsid.

RIVEM should also be cited in the fig 4 legend where mentioned.

How was buried surface determined?

Please include more detail about how three conformational states induced by 2H12 were derived. How did the 2D classes correlate? How was radius determined?

A better structural comparison of S2 and S3 is needed (other than the supplemental figure 8 that is not very informative). What is the RMSD between S2 and S3? If these are in fact different conformations, we need to see the superimposition of the structural proteins and a description of the similarities and differences. Further issues with the interpretation that S2 and S3 are different conformations:

The difference in diameter of 1 Angstrom between S2 and S3 is not significant (not believable) at this resolution.

For accurate genome comparison, the capsid density should be subtracted for a 2D classification. One cannot rely on 3D classification to compare genome density as a few empty particles could be included in the potential A-particle class.

How well does S2 align to A-particle (135S) structure?

It cannot be understood from the micrograph shown in supplemental if there are any empty particles in the data. Perhaps the S2 and S3 classes are mixtures of A-particle and empty particles. Possibly the class corresponding to S2 is mostly comprised of A-particles and the S3 has more 80S empty particles.

In Fig 5 a, b, and c (S1, 2, and 3) are reported to be colored radially according to the same key, but S2 has more expansion (more red showing) than S3. Was the color key altered for C?

Susan Hafenstein

Reviewer #3 (Remarks to the Author):

Review Summary:

The study reports two new EV-D68 neutralizing monoclonal antibodies (mAbs), named 2H12 and 8F12, isolated from antigen immunized mouse hybridomas. As there is no vaccine or effective therapeutic treatment of the virus, the described two mAbs showed potent efficacy targeting two clades (clade B and clade D) of EV-D68 strain when used in combination. Two mAbs showed significant survival improvement using mouse infection model and neutralized two different clades of EV-D68 strain with different potencies in cell culture assay. Cryo-EM structure analysis of the Fab complex with EV-D68 showed a similar epitope binding region for both antibodies.

Although the two antibody cocktail (1:1 ratio) had differential neutralizing activity against two different clades in cocktail, the combined treatment data did not demonstrate synergistic activity. Therefore, the proposed usage of cocktail of two mAbs is not well supported by the data set and there is a limited therapeutic value in consideration of more costly for development of cocktail in comparison with development of single monoclonal antibody.

Major points to be addressed:

- Title: The study did not cover 'development' aspect, should take out 'Development' out the title.
- Rational of using cocktail of two mAbs in the study is not well defined. Authors should run a binding competition assay to dissect whether the two mAbs can compete each other in viral binding.

- Analysis of cryo-EM structures of the two viral/Fab complexes did not clearly point out the differences of the two mAbs in binding epitopes and the mechanisms of action of the two mAbs.
- Need to clarify if the two antibodies work cooperatively or independently as the data suggest that two mAbs neutralize two different clades of EV-D68 and appears as additive in action.
- The study made chimeric (mouse variable/human constant region) version of the two antibodies in the study (Figure 7). Humanized antibodies need to use human germline in variable sequences and commonly do CDR grafting into human germline to reach more than >90% sequences as human origin.

Other points in the data analysis, interpretation and conclusions:

- Figure 1a: Binding affinity data (KD, kon, kdis) should follow after the Octet sensorgrams for easy following and clarity. The data in Figure1a should have standard deviations calculated from minimum of three independent experimental repeats.
- Figure 7b: It is misleading and not conclusive to demonstrate humanization using WB detection with anti-mouse IgG or anti-human IgG. Purity of antibodies and Fab used in structure work should be analyzed using size exclusion (SEC) chromatography to determine other contaminants, aggregates, and purity.

Dear reviewers,

We would like to thank you for your feedback. We have responded to the points raised by all the reviewers and highlighted the changes in yellow in the revised text.

Response to reviewer #1' s comments:

Reviewer #1 (Remarks to the Author):

In this manuscript Zhang, et al detail a study of two murine monoclonal antibodies (mAbs) isolated by the authors after immunization of mice with enterovirus D68 (EV-D68) virus-like particles. These two mAbs have unique neutralization profiles across different clades of EV-D68 despite binding to nearly identical epitopes as identified by the authors using cryo-EM. Further, they performed mechanistic studies to identify how the antibodies neutralize EV-D68, by blocking interactions with sialic acid and inhibiting attachment to cells in vitro. The cryo EM structures also indicate that one of the mAbs, 2H12, may additionally neutralize by capturing a novel structural intermediate of the virus that can allow escape of the genomic material. Finally, they show that both native murine and chimeric human forms of the mAbs can protect mice from EV-D68 challenge when given up to 3 or 1 days after infection, respectively.

The epitopes identified are unique from the only other structurally identified EV-D68 mAb epitopes in the literature: two murine mAbs in reference 19 and two human mAbs in reference 25. The macaque mAb epitope in reference 24 was not confirmed with structural methods. This represents an advance in the understanding of how antibodies can interact with EV-D68, a medically significant virus responsible for causing outbreaks of a severe paralyzing illness, acute flaccid myelitis. The difference in neutralization mechanisms despite nearly overlapping epitopes is also of general interest in the understanding of antibody-virus interactions.

The work in this paper is quite thorough and generally well arranged and easy to follow.

Response: We thank the reviewer for the positive comments. Below are our responses to the reviewer's specific concerns.

Q1-1: There are a number of statements of primacy, though, that are unfounded or unnecessary. This is especially noted with the use of "elite" in line 408 and throughout lines 72-83. For example, the study in reference 25 does in fact test the therapeutic window of giving mAbs at different time points after infection in mice, which is counter to the claims in lines 78-79. Based on this, the use of "unprecedented" in line 455 is not applicable.

A1-1: Thanks for the comments. As suggested, we have toned down the statements

and made modifications accordingly in our revised manuscript. Specially, we **deleted** “However, in the three studies^{19, 24, 25}, therapeutic effectiveness against diverse EV-D68 clades and/or therapeutic window of the anti-EV-D68 MAbs have not been demonstrated, making these MAbs less attractive for further development into therapeutics for future application in real-life clinical settings. Therefore, it is important to continue searching for more powerful anti-EV-D68 MAbs suitable for developing MAb-based therapy for treating EV-D68 infections in human” (previous lines 77-83). In addition, the description “an elite drug candidate” (previous line 408) was modified as “a potential drug candidate”. The statement “unprecedented therapeutic potency and breadth” (previous line 455) was modified as “remarkable therapeutic potency and breadth”.

Q1-2: It is also curious that the discussion in lines 410-431 compares the mAbs in this study to prior referenced murine and macaque mAbs but not the human mAbs, which have equivalent to stronger neutralization potency against the same EV-D68 isolates. In any case, it is difficult to quantitatively compare neutralization potencies across different studies because they are measured using different techniques in different laboratories. The studies in this paper are of sufficient interest and quality that the comparisons are not necessary to bolster the significance.

A1-2: Thanks. We have followed the suggestion to delete the statement about the comparison of neutralization capacity of different antibodies (previous lines 410-431).

Q1-3: The statement in lines 220-222 could be reworded to improve accuracy, as the mAb cocktail was never superior to a single mAb in any of the in vivo experiments. This statement and the use of the word “synergy” in line 159 seem to imply that the antibody cocktail of 2H12 and 8F12 is superior in efficacy to either mAb individually. That would demonstrate synergy. What the authors show is simply that each antibody individually is more effective at neutralizing or protecting against some EV-D68 strains than others. Therefore, giving a cocktail of mAbs provides redundancy, but there is no synergy. The combination of mAbs is no more effective than the most effective individual mAb in the cocktail.

A1-3: We agree with your opinion. We apologize for the unclear descriptions. To avoid confusion, we have modified the manuscript accordingly. The statement “the 2H12/8F12 cocktail is superior to single MAbs in treating diverse EV-D68 infections at a delayed time point” (previous lines 220-222) was modified as follows: “the 2H12/8F12 cocktail is very effective in treating diverse EV-D68 infections at a delayed time point”. In addition, the description “to create synergy” (previous line 159) was deleted.

Q1-4: The authors use the term “humanized” (first used in line 374) in a manner inconsistent with the antibody field. The antibodies used in this study are mouse-human chimeras, or “chimeric” antibodies. The term humanized more specifically implies that only the CDR loops of the murine mAb were used in place of the CDRs of a human mAb backbone. The authors swapped out the entire variable regions, though, which is not truly “humanization.” This has important clinical implications because truly humanized mAbs are less immunogenic to humans than the mouse-human chimeras described in this manuscript. Variations of the word humanized should be eliminated from the manuscript and replaced with variations of the word chimeric.

-Description of how palivizumab was humanized, swapping out only the CDRs: S Johnson et al, J Infect Dis, 1998; PMID: 9359721 DOI: 10.1086/514115
-Description of chimeric antibodies: <https://absoluteantibody.com/our-technology/formats-we-have-made/chimeric-antibodies/>

-Description of how palivizumab was humanized, swapping out only the CDRs: S Johnson et al, J Infect Dis, 1998; PMID: 9359721 DOI: 10.1086/514115

A1-4: Thanks for pointing this out. As suggested, the word “humanized” has been replaced with the words “human-mouse chimeric” or “chimeric” in our revised manuscript.

Q1-5: The authors should further clarify that the specific antibodies they use for the western blot in Fig. 7b, discussed in lines 380-382, are Fc-specific. This is not mentioned in the methods or in the main text. One would otherwise expect a polyclonal anti-mouse IgG antibody to react to both the native murine mAb and the human-chimeric mAb.

A1-5: Thanks for the suggestion. We have modified the manuscript accordingly (please see lines 436-438), to read “Chimeric MAbs 2H12 (c2H12) and 8F12 (c8F12) reacted with anti-human IgG (**constant domains**) antibody but not anti-mouse IgG (**constant domains**) antibody in western blot assays”. In addition, the statement “Anti-human IgG or anti-mouse IgG secondary antibodies (HRP conjugate) used in this study recognize the constant domains of both heavy and light chains” has been included in Fig. 7b legend.

Q1-6: The authors frequently refer to reference 32 (lines 401 and 432) in describing a 1-3 day incubation period for EV-D68 in humans. This reference is a small study that specifically refers to respiratory illness. The therapeutic mouse models described in this manuscript focus on a neurologic phenotype (paralysis) related to EV-D68 infection. Much larger studies have described median 5 days of illness preceding limb weakness in AFM, the typical human neurologic phenotype of EV-D68 infection. This longer period of incubation to AFM should be acknowledged in the manuscript.

-Study showing 5 day incubation: J Lopez et al, MMWR 2019, PMID: 31295232 DOI: 10.15585/mmwr.mm6827e1

A1-6: Thanks for the constructive comments. As suggested, we cited the paper “Vital Signs Surveillance for Acute Flaccid Myelitis — United States, 2018” and modified the statements (lines 458-460), to read “a prodromal illness (such as febrile, respiratory, and/or gastrointestinal) preceded the onset of limb weakness in most AFM patients by a median of 5 days^{32, 33}” (Ref 32. Lopez A, et al. Vital Signs: Surveillance for Acute Flaccid Myelitis - United States, 2018. MMWR Morb Mortal Wkly Rep 68, 608-614 (2019); ref 33. Messacar K, et al. Acute flaccid myelitis: A clinical review of US cases 2012-2015. Ann Neurol 80, 326-338 (2016)).

Q1-7: There are a few graphs in the paper that appear to be lacking error bars, specifically the ELISA in Fig. s2 and the clinical scores in Figs. 2, S3B, and 7H. Regarding the clinical scores, it is impossible to understand the heterogeneity in clinical disease in the mice without them. Alternatively, the authors could create a supplemental table listing the clinical scores for each mouse to give an idea of this heterogeneity.

A1-7: Thanks for the constructive comments. There are error bars in Fig. S2, but they are too small to be visible. In addition, as suggested, we have remade the graphs (Figs. 2, S3B, and 7H) with error bars, by entering raw data (clinical scores of each mouse) instead of averaged data in the GraphPad Prism software.

Fig. 2 Therapeutic efficacy of the MAbs against EV-D68 infection in mice. One-day-old ICR mice ($n = 13\text{--}27/\text{group}$) were inoculated intraperitoneally (i.p.) with strain 18947 **a-b** or strain 18953 **c**. The suckling mice were i.p. injected with PBS, 10 $\mu\text{g/g}$ of 2H12, 10 $\mu\text{g/g}$ of 8F12, or a mixture of both MAbs (10 $\mu\text{g/g}$ of each MAb) at 1 day post-infection (dpi) **a** or 3 dpi **b, c** and were then monitored daily for survival and clinical score. Clinical scores were graded as follows: 0, healthy; 1, lethargy and reduced mobility; 2, limb weakness; 3, limb paralysis; 4, death. Survival rates of antibody-treated mice were compared with the mice in the PBS control group. Statistical significance was indicated as follows: ns., no significant difference ($P \geq 0.05$); *, $P < 0.05$; ***, $P < 0.001$. Note that to prevent overlap, the overlapping data sets in left panels (survival curves) were nudged by 5 units in the Y direction. All error bars represent SEM.

Fig. 7. (g, h) Therapeutic efficacy of chimeric MAbs against EV-D68 infection in neonatal mice.

One-day-old ICR mice (n = 14–15/group) were inoculated i.p. with strain 18947. The suckling mice were i.p. injected with PBS, 10 µg/g of c2H12, or 10 µg/g of c8F12 at 1 dpi. Clinical scores were graded described in Figure 2 legend. Survival of mice in each antibody-treated groups was compared to the PBS control group. Statistical significance was indicated as follows: ***, P < 0.001. Error bars represent SEM.

Supplementary Figure 3. Prophylactic efficacy of MAbs 2H12 and 8F12 against EV-D68 infection in neonatal mice. Groups of ICR mice (age < 24h; n = 12–14/group) were i.p. injected with PBS, 10 µg/g of 2H12, 8F12, IgG2a isotype control (ctr) MAb (1C11), or IgG2b isotype control MAb (1F4) and one day later challenged with strain 18947. The challenged mice were monitored daily for (a) survival and (b) clinical score. Clinical scores were graded as follows: 0, healthy; 1, lethargy and reduced mobility; 2, limb weakness; 3, limb paralysis; 4, death. Survival of mice in each antibody-treated groups was compared to the PBS control group. Statistical significance was indicated as follows: ns., no significant difference (P ≥ 0.05); **, P < 0.01; ***, P < 0.001. Error bars represent SEM.

Response to reviewer #2' s comments:

Reviewer #2 (Comments to the Author):

This is an important work to characterize EV-D68 neutralizing Mabs, 2H12 and 8F12. Their efficacy as a cocktail for treatment of infections is explored by multiple assays including a structural characterization. The largest concern is a lack of careful quantification to accompany the structural analysis. There are a few other minor concerns, necessary clarifications, and areas where more information is needed.

Response: We thank the reviewer for the positive comments on our work.

Q2-1: Line 93 and elsewhere, 2.9 is not atomic resolution and should be described as high resolution or near atomic resolution.

A2-1: The point is well taken. We have modified related descriptions accordingly in our revised manuscript (please see lines 91, 275, and 503), to read “The high resolution (up to 2.9 Å) cryo-electron microscopy (cryo-EM) structures”, “illustrating the high-resolution of the map”, and “Importantly, our high-resolution cryo-EM structures revealed”.

Q2-2: Line 142 Bio-layer interferometry (BLI) is not defined or cited when first used in the Results.

A2-2: Thanks for pointing this out. As suggested, we have defined this term in our revised manuscript, to read “determine the binding affinity of the MAbs to the 18947 virion by bio-layer interferometry (BLI)” (please see line 142).

Q2-3: Manuscript would greatly benefit from a table listing the different strains and each source and abbreviation used throughout.

A2-3: Thanks for the constructive suggestion from the reviewer. We have now added a table to summarize the information of EV-D68 strains used in our study as Supplementary Table 1 in our reviewed manuscript. For the convenience of the reviewers, we also show it here.

Supplementary Table 1. A summary of all the EV-D68 strains used in this study.

EV-D68 strains	Abbreviation	Clade	Source (catalog)	Genbank ID
US/MO/14-18947	18947	B	ATCC (VR-1823)	KM851225
US/MO/14-18950 ^a	18950	B	P1 gene was synthesized	KM851228
US/KY/14-18953	18953	D	ATCC (VR-1825)	KM851231
Fermon	Fermon	Prototype	ATCC (VR-1826)	AY426531

^a VLP vaccine strain.

Q2-4: There is not enough description for the cryo EM section. Description of the unchanged virus capsid should include quantification. What is the RMSD of the capsid

superimposed with virus versus A-particle structures? Such a comparison should be the basis of a conclusion that the FAB does not induce change, not a visual inspection and speculation that the two fold is closed or open.

A2-4: The point is well taken. We have calculated the RMSD values of the capsid proteins between our structures and the superimposed full native or A-particle of the EV-D68 structure, which is now provided as Supplementary Table 3 in our revised manuscript. The RMSDs are calculated based on the maximum shared C α atoms of protomers in each pair of molecules excluding the VP4 subunit. We have also modified the text accordingly, to read “The capsid structure in the EV-D68/8F12 complex is essentially identical to that of the unbound native EV-D68 virion¹⁸ with the overall C α root-mean-square deviation (RMSD) value being rather small (0.53 Å) between the protomers of the two structures (Supplementary Table 3); also, the two-fold axis channel remains closed in the complex (Fig. 4c). Collectively, these data indicate that 8F12 Fab binding does not induce obvious conformational changes of the EV-D68 virion.”, and “In addition, we compared our structures with the available cryo-EM structures of the native mature virion (PDB: 6CSG), expanded 1 (E1, PDB: 6CS3), and A particle (PDB: 6CS6) of the 18947 strain¹⁸ (Supplementary Table 3). The overall C α RMSD between the capsid protomer of the 18947 virion and that of our S1 state was calculated to be as small as 0.86 Å, indicating that the viral particle in S1 adopts the native conformation. A better fit in capsid protomer structure was observed between our S2 state and the 18947 E1 particle (RMSD = 1.21 Å) than between S2 and the 18947 native virion (RMSD: 1.99 Å) or between S2 and A-particle (RMSD: 3.52 Å), suggesting that S2 is more close to the E1 conformation instead of resembling the A-particle. Our S3 state fits better with the 18947 A-particle in the capsid protomer (RMSD: 1.61 Å) than with the other two structures (mature virion and E1 particle), but with significantly weaker RNA genome density (Fig. 5i) than typical A-particle (with genome RNA).” (please see lines 282-288 and 349-361).

Supplementary Table 3. RMSD values between capsid protomers.

RMSD (Å)	EV-D68/8F12	EV-D68/2H12 (S1)	EV-D68/2H12 (S2)	EV-D68/2H12 (S3)
Mature EV-D68 (PDB: 6CSG)	0.53	0.86	1.99	2.51
EV-D68 A-particle (PDB: 6CS6)	4.09	4.20	3.52	1.61
EV-D68 E1 particle (PDB: 6CS3)			1.21	1.86

Q2-5: Line 316, in all reports to data, genome release at the two fold is suggested.

This is a model not proven yet, but merely suggested by numerous A-particle structures.

A2-5: Thanks for the suggestion. We have modified the related statement, to read “Noteworthy, enterovirus genomes have been **suggested** to be released through the two-fold axis channel^{30, 31}”.

Q2-6: Include a central section for 8F12 complex map as well so that the quality of the density can be seen.

A2-6: The suggestion is well taken. We have now included this central section of EV-D68/8F12 complex map in the revised Supplementary Fig. 4e (we also show it below for the convenience of the reviewer) and added related description in manuscript, to read “We also show the central section of the EV-D68/8F12 map to illustrate the quality of the densities corresponding to the viral capsid and Fab (Supplementary Fig. 4e).” in lines 276-277.

Supplement Figure 4e. Central section for EV-D68/8F12 map.

to reporting on the orientation of the light and heavy chains relative to the capsid.

A2-7: We followed the suggestion from our reviewer to make the description for model building of Fab more clear by adding the following sentence “Here only models of the variable regions of the Fab were built, since electron densities corresponding to the constant regions were relatively weak (Supplementary Fig. 4d, more details for model building were described in the Methods).” (please see lines 278-280).

Q2-8: RIVEM should also be cited in the fig 4 legend where mentioned.

A2-8: The suggestion is well taken, and we have now added the citation for RIVEM in Fig.4 legend, to read “generated using RIVEM (Radial Interpretation of Viral Electron density Maps; reference [62])” .

Q2-9: How was buried surface determined?

A2-9: We used the PISA server (https://www.ebi.ac.uk/pdbe/prot_int/pistart.html)¹ to determine the buried surface, which is a commonly used software for this purpose². We first uploaded the coordinate file containing a protomer and the associated Fab,

then the PISA server could calculate the interface area between chains, which is an average of solvent-accessible area in interface from the two calculated chains. We have modified the related methods and Supplementary Table 6 in the revised manuscript, to read “Fab-virion interaction analysis including hydrogen bond, salt bridge prediction, **and buried surface area calculation** were carried out through PISA server (https://www.ebi.ac.uk/pdbe/prot_int/pistart.html)”, and “Surface area of EV-D68 buried by MAb 8F12 and 2H12 (in state S1) determined by the PISA server.” (please see lines 774-776 and Supplementary Table 6).

Q2-10: Please include more detail about how three conformational states induced by 2H12 were derived. How did the 2D classes correlate? How was radius determined?

A2-10: The suggestion is well taken. We have now added more details on how the three conformational states induced by 2H12 were derived in the updated Methods section, to read “Similar processing tactics were applied to EV-D68/2H12 dataset (Supplementary Fig. 6). After 2D classification, 8,844 particles remained. After CTF refinement and Bayesian polishing, these particles were reconstructed into a 3.25 Å resolution map. Further 3D classification of this dataset into three classes revealed two distinct major conformations with better structural features (class 1 and 3, Supplementary Fig. 6a). Class 1 map depicts the full native virion feature and no other conformation could be detected after further 3D classification, which was thus termed S1 state. Meanwhile, class 3 map exhibits open channels at the two-fold axis, a characteristic feature of expanded enteroviral particles. We then performed further 3D classification on this dataset and obtained two distinct conformational states (termed S2 and S3, Supplementary Fig. 6a). Compared with S2, S3 shows different internal genomic RNA organization and significantly weaker RNA genome density. Although the overall capsid structures of S2 and S3 resemble each other, there are still some differences, for example, density for VP4 is totally missing in S3, which is not the case in S2.” (please see lines 744-757).

For the convenience of our reviewers, we also show Supplementary Fig. 6a below.)

Supplementary Figure 6. Cryo-EM structural analysis of EV-D68/2H12 complex. (a) Representative cryo-EM image of the EV-D68/2H12 complex and the flowchart of the 3D reconstruction process for the complex. Full particle, partially solid-core particle, and empty particle are marked in red, yellow, and green boxes, respectively. Yellow arrow indicates the broken particle induced by 2H12 binding and the exposed viral genome.

We computationally sorted out these three states in the EV-D68/2H12 dataset, and some of the subtle feature differences such as the open channels at the two-fold axis is hard to be directly visualized in the reference-free 2D class averages. Still, for the overall genome occupancy status, the 2D analysis displays different characteristics, for instance, S1 showed representative characteristics of full native virion with strong RNA content, the RNA content density of S2 is comparable to S1, while for S3, the internal genome density was significantly weaker than that of S1 and S2 (Fig. R1). These 2D average characteristics are consistent with the observations in the related 3D reconstructions.

Fig. R1 Representative reference-free 2D class averages of (a) S1, (b) S2, and (c) S3 states for the EV-D68/2H12 complex. Density for inner RNA genome is weaker in S3 than the other two states.

For the radius measurement in Fig. S8, we first measured the diameter of the virus capsid through measuring the distance between the CG2 atom in residue V231 of VP1 from protomers located in the opposite direction on the diagonal, then divided the diameter by two to deduce the radius value. This atom is located at the most protruding part of the virus capsid. We have included this information in the legend of Supplementary Fig. 8.

Q2-11: A better structural comparison of S2 and S3 is needed (other than the supplemental figure 8 that is not very informative). What is the RMSD between S2 and S3? If these are in fact different conformations, we need to see the superimposition of the structural proteins and a description of the similarities and differences. Further issues with the interpretation that S2 and S3 are different conformations. The difference in diameter of 1 Angstrom between S2 and S3 is not significant (not believable) at this resolution.

A2-11: Thanks for the constructive suggestion from the reviewer. We have now provided more detailed structural comparison between S2 and S3 states (new Supplementary Fig. 8b-8e), to read "Further structural comparison between the S2 and S3 states, by superimposition of their protomer structures together (Supplementary Fig. 8b-e), revealed the overall C α RMSD value between the two structures to be 1.47 Å, indicating conformational variations between the two states. Specifically, the N-terminus of VP1 is externalized and the first 40 residues are

missing in S3, while most of them could be resolved lying underneath the VP3 in S2; the VP1 C-terminal loop (residues K270 to T289) is largely missing in S3, while captured in S2 (Supplementary Fig. 8c). Noteworthy, VP4 is completely missing in S3, while part of VP4 is present in S2 structure (Supplementary Fig. 8b, 8d). Moreover, residues P43 to T54 from VP2 AB loop are also missing in S3 but captured in S2 (Supplementary Fig. 8d); residues P168 to T174 within VP3 GH loop display significantly different conformations in the two states (Supplementary Fig. 8e). Collectively, S2 and S3 are indeed in different states in capsid conformation.” (see lines 337-348).

Supplementary Figure 8. (b-e) Structural comparison between the EV-D68/2H12 S2 (color) and S3 (gray) states for the protomer **(b)** and individual subunit **(c-e)**. The major conformational differences between the two states are indicated by arrows.

We agree with this reviewer that the difference in diameter of 1 Å between S2 and S3 is not significant. We just listed the number we measured to address that both S2 and S3 were expanded compared to S1 and didn't imply there was any notable difference in diameter between S2 and S3. We should emphasize that the main differences between S2 and S3 lie in the capsid protein composition, conformation, and RNA density as mentioned above.

Q2-12: For accurate genome comparison, the capsid density should be subtracted for a 2D classification. One cannot rely on 3D classification to compare genome density as a few empty particles could be included in the potential A-particle class.

A2-12: Thanks for the suggestion from our reviewer. We agree that 3D classification may be not enough and 2D classification on subtracted particles is a good approach. As suggested, we subtracted the capsid density of virion particles in the EV-D68/2H12 S2 and S3 datasets, and performed reference-free 2D classification with a mask of 300 Å in diameter to further exclude the influence of capsid. All the 2D class averages have inner content and no obviously empty particle classes were found as concerned. The resulting capsid-subtracted 2D class averages of S2 and S3 datasets show different inner genome organization (Fig. R2). For instance, the inner genomic RNA appears full and tightly packed in S2, while that in S3 appears sparse and less dense. These characteristics indicate that S3 possesses less genome content than S2. This conclusion is consistent with our 3D classification results.

Fig. R2 Reference-free 2D class averages of capsid-subtracted particles for EV-D68/2H12 S2 (a) and S3 (b) datasets.

Q2-13: How well does S2 align to A-particle (135S) structure?

A2-13: We should point out that S2 resembles the expanded 1 (E1) state, but not A particle, as stated in our initial manuscript and further substantiated by RMSD analysis (Supplementary Table 3). Specially, C α RMSD between S2 and A-particle (PDB: 6CS6) protomer is 3.52 Å, while C α RMSD between S2 and E1 (PDB: 6CS3) is reduced to 1.21 Å.

Still, we followed the suggestion from the reviewer to align S2 with A-particle (Fig. R3), which reveals that there are multiple conformational variances between the two states. In state S2, the N-terminus of VP1 and VP4 is ordered and located inside the

capsid shell; whereas in A particle, the N-terminal tail of VP1 is externalized and VP4 is missing (expelled from the capsid)³. These main characteristics distinguish S2 from A-particle. Besides, there are other obvious differences. For instance, the C-terminal residues K270 to T289 of VP1 are missing in A-particle but resolved in S2; there is a shift in VP1 β -barrel; VP2 AB loop, VP2 C-terminus, and VP3 GH-loop also show significant conformational differences between S2 and A-particle (Fig. R3). Collectively, these data demonstrate that viral capsid of S2 is different from that of A-particle.

Fig. R3 Structural comparison of the protomers (a) and individual subunit proteins (b-d) of the EV-D68/2H12 S2 (color) and A-particle (gray) states. The major structural differences between the two structures are indicated by arrows.

Q2-14: It cannot be understood from the micrograph shown in supplemental if there are any empty particles in the data. Perhaps the S2 and S3 classes are mixtures of A-particle and empty particles. Possibly the class corresponding to S2 is mostly comprised of A-particles and the S3 has more 80S empty particles.

A2-14: As shown in our raw micrographs (Fig. R4a), the majority of the particles in our sample are particles with genomes, and there are seldom empty particles. We only chose genome RNA-containing particles for subsequent 3D reconstruction. Moreover, we run multiple rounds of 2D classification to exclude the empty particles. 2D classification of the remaining 8,844 particles used for final structural determination also confirmed that there is no empty particles in the dataset (Fig. R4b). Therefore,

the particles used for 3D construction are all RNA-filled particles. Therefore, S3 (new intermediate with weak RNA density) is different from 80S empty particles (no RNA).

As mentioned above, viral capsid in S2 is different from that in A-particle (please see A2-13).

Fig. R4. Inspection of the EV-D68/2H12 dataset. **(a)** Raw Cryo-EM micrographs of the EV-D68/2H12 complex. Bar = 50 nm. Full particle, partially solid-core particle, and empty particles are marked in red, yellow, and green boxes, respectively. Yellow arrow indicates the broken particle induced by 2H12 binding and the exposed viral genome. **(b)** 2D class averages of the remaining 8,844 particles used for final structural determination for EV-D68/2H12 complex.

Q2-15: In Fig 5 a, b, and c (S1, 2, and 3) are reported to be colored radially according to the same key, but S2 has more expansion (more red showing) than S3. Was the color key altered for C?

A2-15: The color key remains the same for Fig. 5a, b, and c. Here the red color represents smaller radius within 100 Å, corresponding to the enclosed interior genome density, instead of illustrating expansion. The S2 shows more inner red density than S3 is because though S2 and S3 has the same level of expansion, S2 has more genomic RNA density in its inner space than that of S3 (Fig. 5 h-i, and Fig. R2).

Response to reviewer #3' s comments:

Reviewer #3 (Comments to the Author):

Review Summary:

The study reports two new EV-D68 neutralizing monoclonal antibodies (mAbs), named 2H12 and 8F12, isolated from antigen immunized mouse hybridomas. As there is no vaccine or effective therapeutic treatment of the virus, the described two mAbs showed potent efficacy targeting two clades (clade B and clade D) of EV-D68 strain when used in combination. Two mAbs showed significant survival improvement using mouse infection model and neutralized two different clades of EV-D68 strain with different potencies in cell culture assay. Cryo-EM structure analysis of the Fab complex with EV-D68 showed a similar epitope binding region for both antibodies.

Q3-1: Although the two antibody cocktail (1:1 ratio) had differential neutralizing activity against two different clades in cocktail, the combined treatment data did not demonstrate synergistic activity. Therefore, the proposed usage of cocktail of two mAbs is not well supported by the data set and there is a limited therapeutic value in consideration of more costly for development of cocktail in comparison with development of single monoclonal antibody.

A3-1: We agree with the reviewer that the combined treatment data did not demonstrate synergistic activity. However, we need to emphasize the fact that the 2H12/8F12 cocktail conferred broader neutralization and *in vivo* protection than single MAbs (Fig. 1a and 2), supporting the notion that use of antibody cocktail is a better option for neutralizing and treating diverse EV-D68 clades than the use of a single antibody.

Antibody cocktail has been used in the treatment of viral infections. For example, Regeneron's antibody cocktail REGN-EB3 (Inmazeb®) is a FDA-approved treatment for Ebola and please note that REGN-EB3 is a cocktail of three monoclonal antibodies, REGN3470, 3471, and 3479. In addition, the two antibodies (2H12 and 8F12) could be made into a bispecific antibody, the cost of which is comparable to the cost of a single antibody.

Q3-2: Major points to be addressed:

- Title: The study did not cover 'development' aspect, should take out 'Development' out the title.

A3-2: Thanks for the comment. As suggested, the title has been changed to "Functional and structural characterization of a two-MAb cocktail for delayed treatment of enterovirus D68 infections".

Q3-3: Rational of using cocktail of two mAbs in the study is not well defined. Authors

should run a binding competition assay to dissect whether the two mAbs can compete each other in viral binding.

A3-3: Rationale for using cocktail of two mAbs was mentioned in our study, to read “The distinct cross-neutralization profiles of 2H12 and 8F12 propelled us to investigate whether they can be used in combination” (please lines 160-162). MAb 2H12 and 8F12 displayed distinct cross-neutralization profiles. Specially, 2H12 potently neutralized the clade D strain 18953 but almost had no neutralization effect on the prototype strain Fermon whereas 8F12 efficiently neutralized Fermon but poorly neutralized strain 18953 (Fig. 1a). By contrast, the 2H12/8F12 cocktail potently neutralized all tested EV-D68 strains (Fig. 1a). Thus, compared with single MAbs, antibody cocktail showed broader neutralizing effects. To emphasize this point, we have modified the statements in our revised manuscript (please see lines 159-160, 166-168), to read “The distinct cross-neutralization profiles of 2H12 and 8F12 propelled us to investigate whether they can be used in combination **to improve neutralization breadth.**” and the conclusion “In another word, neutralization breadth was improved by combining the 2H12 and 8F12 antibodies, providing a strong rationale for using the cocktail but not single MAbs for pan-EV-D68 neutralization”.

As suggested, a competitive binding assay using BLI was performed in order to examine whether MAb 2H12 would compete with MAb 8F12 for viral binding. In this assay, immobilized EV-D68 virion was saturated with the first antibody 2H12 and then allowed to bind the second antibody 8F12. As shown in the Fig. 6e, the initial binding of 2H12 to EV-D68 virions blocked subsequent 8F12 binding. Collectively, our structural and biochemical data demonstrate that the binding epitopes of 2H12 and 8F12 are similar or largely overlapping (Fig. 6f).

Fig. 6. Comparison of the two immune complex structures. **e** A competitive binding assay using BLI. Immobilized EV-D68 virion (18947 strain) was saturated with the first antibody 2H12 and then allowed to react with 2H12 alone (control) or the antibody mixture (8F12 + 2H12).

Q3-4: Analysis of cryo-EM structures of the two viral/Fab complexes did not clearly

point out the differences of the two mAbs in binding epitopes and the mechanisms of action of the two mAbs.

A3-4: Thanks for the comment. In general, the binding footprints of 2H12 and 8F12 on the virion surface are very similar except that 8F12 can also interact with the VP1 BC loop (Fig. 4 and 5), suggesting that the two antibodies may target the same antigenic site. To verify this structure-based prediction, we performed a BLI competitive binding assay, in which immobilized EV-D68 virion was saturated with the first antibody 2H12 and then allowed to bind the second antibody 8F12. As shown in the Fig. 6e, initial binding of 2H12 to the EV-D68 virions blocked subsequent 8F12 binding. Hence, both structural and biochemical data demonstrate that the binding epitopes of 2H12 and 8F12 are similar or largely overlapping (Fig. 6f).

Both 2H12- and 8F12-binding footprints overlap with the sialic acid receptor binding site (Fig. 4d, 4f, 5j, 5l) and both MAbs can inhibit the binding of EV-D68 to sialic acid receptors in HI assays (Fig. 3), indicating that blockade of EV-D68 binding to sialic acid receptors via steric hindrance is the main neutralization mechanism shared by MAbs 2H12 and 8F12 (Fig. 6f). In addition, our structural data showed that 2H12 binding could destroy viral particles to some extent (Supplementary Fig. 6a) and trigger premature virus uncoating (Fig. 5g-i), likely leading to impaired virus infectivity. Therefore, MAb 2H12 may adopt two additional mechanisms to achieve neutralization, including impairing virion integrity and inducing premature virion uncoating (Fig. 6f).

To make our point clear, we have now added the summary of the epitopes and neutralization mechanisms to Fig. 6f and modified the manuscript accordingly (please see lines 409-426).

MAb	Epitope	Neutralization mechanism
2H12	VP1-GH loop	1. Blocking virus-receptor interactions
	VP2-EF loop	2. Impairing viral particles
	VP3-C terminus	3. Inducing premature virus uncoating
8F12	VP1-BC loop	1. Blocking virus-receptor interactions
	VP1-GH loop	
	VP2-EF loop	
	VP3-C terminus	

Fig. 6. Comparison of the two immune complex structures. f Summary of the epitopes and neutralization mechanisms of the 2H12 and 8F12 MAbs.

Q3-5: Need to clarify if the two antibodies work cooperatively or independently as the data suggest that two mAbs neutralize two different clades of EV-D68 and appears as additive in action.

A3-5: Our biochemical and structural data indicate that the 2H12 and 8F12 MAbs work independently but not cooperatively. Cross-neutralization assay indicated that

the antibody combination (2H12/8F12) is no more effective than the most effective individual MAb (2H12 for strain 18953; 8F12 for strain Fermon) (Fig. 1a), suggesting that there is no synergy when the two MAbs were combined. To emphasize this point, we have modified the manuscript accordingly, to read “It is likely that 2H12 and 8F12 act independently and exert preferential neutralization on different virus strains, as the 2H12/8F12 combination was no more effective than the most effective individual MAb for a given strain (eg. 2H12 for strain 18953 and 8F12 for strain Fermon) (Fig. 1a). However, the two MAbs in the cocktail nicely complement each other to achieve broad neutralization against diverse virus strains both in vitro and in vivo (Fig. 1a and 2), thus highlighting the advantage of the 2H12/8F12 antibody cocktail.” (please see lines 473-479).

Q3-6: The study made chimeric (mouse variable/human constant region) version of the two antibodies in the study (Figure 7). Humanized antibodies need to use human germline in variable sequences and commonly do CDR grafting into human germline to reach more than >90% sequences as human origin.

A3-6: Thanks for pointing this out. As suggested, the word “humanized” has been replaced with the words “human-mouse chimeric” or “chimeric” in our revised manuscript.

Other points in the data analysis, interpretation and conclusions:

Q3-7: Figure 1a: Binding affinity data (KD, kon, kdis) should follow after the Octet sensorgrams for easy following and clarity. The data in Figure1a should have standard deviations calculated from minimum of three independent experimental repeats.

A3-7: Thanks for the suggestions. We have put together the binding affinity data and the Octet sensorgrams (please see the new Fig. 1d-e). In addition, the cross-neutralization experiments were repeated several times, and the neutralization concentration values of the MAbs were the same.

Fig. 1 Neutralization activity and binding properties of the MAbs. **d, e** Binding affinities of the MAbs to EV-D68 18947 virion measured by BLI. MAb concentrations used and values of K_D , K_{on} and K_{dis} were shown.

Q3-8: Figure 7b: It is misleading and not conclusive to demonstrate humanization using WB detection with anti-mouse IgG or anti-human IgG. Purity of antibodies and Fab used in structure work should be analyzed using size exclusion (SEC) chromatography to determine other contaminants, aggregates, and purity.

A3-8: Thanks for the comments. Anti-human IgG or anti-mouse IgG secondary antibodies (HRP conjugate) used in this study recognize the constant domains of both heavy and light chains. Chimeric MAbs 2H12 (c2H12) and 8F12 (c8F12) reacted with anti-human IgG (constant domains) antibody but not anti-mouse IgG (constant domains) antibody in western blot assays (Fig. 7b), confirming their chimeric nature. In addition, sequencing analysis of the plasmids used for chimeric antibody expression confirmed that the variable regions of the antibody are of murine origin and the constant regions are of human origin. These data are sufficient to demonstrate chimerization of the antibodies.

As suggested, we use size exclusion chromatography to determine the purity of the recombinant chimeric MAbs. We have added this result and modified the manuscript accordingly, to read “The purified c2H12 and c8F12 were > 94% purity as determined by size exclusion chromatography (Supplementary Figure 9)”

Supplementary Figure 9. The purity of the recombinant MAbs c2H12 (a) and c8F12 (b) was determined by size exclusion chromatography. The retention time and area of all the elution peaks are shown in the inset. The third elution peak of c2H12 and the second peak of c8F12 represent antibody monomer, whereas the first and second peaks of c2H12 and the first peak of c8F12 account for antibody aggregate and/or dimer.

References

1. Xiao C, Rossmann MG. Interpretation of electron density with stereographic roadmap projections. *J Struct Biol* **158**, 182-187 (2007).
2. Zupa E, *et al.* The cryo-EM structure of a gamma-TuSC elucidates architecture and regulation of minimal microtubule nucleation systems. *Nat Commun* **11**, 5705 (2020).
3. Liu Y, Sheng J, van Vliet ALW, Buda G, van Kuppeveld FJM, Rossmann MG. Molecular basis for the acid-initiated uncoating of human enterovirus D68. *Proc Natl*

Acad Sci U S A **115**, E12209-E12217 (2018).

REVIEWER COMMENTS

Reviewer #1 (Remarks to the Author):

The authors have returned a revised manuscript that is greatly strengthened by additional clarifications of their structural studies, rephrasing of key concepts around chimerism vs. humanization and synergy vs. additive effects, and addition of suggested experiments. I have the following minor comments:

In line 431 the authors still use the word "humanized." They should either clarify that the experiments in this manuscript are intended to give credence to the idea that humanization is a worthy effort, or they should drop the use of the term humanized. Because as stated, this implies that the experiments in this paper will determine whether humanized MAb are efficacious, and that is not the case.

In the methods the authors still need to specify that the antibodies used in the western blots are constant domain-specific. They provide a level of detail in their response letter in the beginning portion of A3-8 that will help a general reader and could essentially be copied into the methods.

The labeling of the x-axis for ELISA data in Fig. 1c. and S2 is atypical. The authors show total antibody amounts (ng/well) rather than concentration of antibody (such as ng/mL or $\mu\text{g/mL}$) when displaying ELISA binding data. It would be more typical to label with antibody concentrations, and this would be easy enough to convert. This will also make presentation of the ELISA data the same as the neutralization and HAI data, in which concentration is used rather than amount of antibody.

Just a suggestion, Fig. 7g could benefit from the use of the same arrow convention to indicate time of mAb administration that is used in Fig. 2a-c.

I would appreciate if the authors could clarify their thoughts behind the design of the BLI competition assay depicted in Fig. 6e. I am accustomed to such assays having a few more experimental conditions, including the reciprocal blocking of saturating with 8F12 and then following with 2H12. Further, I am not accustomed to seeing the second antibody condition as a mixture. In other words, I would expect after saturating with 2H12 to then follow by dipping the sensors in a well containing only 8F12, not a combination of 2H12 and 8F12. Since extent of competition requires comparison to the level of binding of the 2nd mAb in the absence of the 1st mAb, binding of the 8F12 mAb alone should be shown alongside the 2H12 and 8F12 competition binding curve to allow the reader to interpret the extent of competition. Additionally, I have typically seen the binding curves corrected so that they normalize to the time of the addition of the second mAb, which is a function that the Octet software can perform. My comments aren't meant to invalidate this experiment, but to raise questions of why the experiment was designed as it was. Even though the conclusion that these mAbs have similar or largely overlapping epitopes is well made overall throughout the manuscript, certainly the reciprocal competition data would strengthen this conclusion and would have been easy to do alongside the included experiments, so is curious by its omission.

As another question of clarification, how did the authors determine which amino acids to label as the sialic acid binding site in orange in Fig. 4f and 5l? Did the authors just use the contact amino acids listed in Table S3 of reference 21? In looking at reference 21, visually the sialic acid binding site outlined in Fig. 2 and S2 of reference 21 has a different appearance than the sialic acid binding site labeled in Fig. 4f and 5l of this manuscript. Could the authors provide this clarification and an explanation if one is known about why these appear different?

Reviewer #2 (Remarks to the Author):

All of my concerns have been addressed.

Reviewer #3 (Remarks to the Author):

The revised manuscript and point to point responses to my questions are adequately addressed.

Dear reviewer,

We would like to thank you for your feedback. We have responded to the points raised by the reviewer and highlighted the changes in yellow in the revised text.

Response to reviewer #1' s comments:

Reviewer #1 (Remarks to the Author):

The authors have returned a revised manuscript that is greatly strengthened by additional clarifications of their structural studies, rephrasing of key concepts around chimerism vs. humanization and synergy vs. additive effects, and addition of suggested experiments. I have the following minor comments:

Response: We thank the reviewer for the positive comments. Below are our responses to the reviewer's specific concerns.

Q1-1: In line 431 the authors still use the word "humanized." They should either clarify that the experiments in this manuscript are intended to give credence to the idea that humanization is a worthy effort, or they should drop the use of the term humanized. Because as stated, this implies that the experiments in this paper will determine whether humanized MAbs are efficacious, and that is not the case.

A1-1: Thanks for the comments. As suggested, the word "humanized" in line 431 has been replaced with the word "chimerized" in our revised manuscript, to read "we then asked whether these two murine MAbs could be chimerized with human IgG constant regions".

Q1-2: In the methods the authors still need to specify that the antibodies used in the western blots are constant domain-specific. They provide a level of detail in their response letter in the beginning portion of A3-8 that will help a general reader and could essentially be copied into the methods.

A1-2: Thanks for the suggestions. We have modified the Methods section to include this description, to read "Western blotting of chimeric MAbs was carried out as described previously ⁶³, except that HRP-conjugated goat anti-mouse IgG (Sigma, USA) and HRP-conjugated rabbit anti-human IgG (Abcam, USA) were used for detection. Note that anti-human IgG or anti-mouse IgG secondary antibodies (HRP conjugate) used in this study recognize the constant domains of both heavy and light chains." (please see lines 796-798).

Q1-3: The labeling of the x-axis for ELISA data in Fig. 1c. and S2 is atypical. The authors show total antibody amounts (ng/well) rather than concentration of antibody

(such as ng/mL or $\mu\text{g/mL}$) when displaying ELISA binding data. It would be more typical to label with antibody concentrations, and this would be easy enough to convert. This will also make presentation of the ELISA data the same as the neutralization and HAI data, in which concentration is used rather than amount of antibody.

A1-3: The point is well taken. As suggested, we have modified the labeling of the x-axis for ELISA data in Fig. 1c. and S2 and have shown the antibody concentration instead of antibody amount. For the convenience of the reviewer, we also show the figures here.

Fig. 1 c Reactivities of the MABs towards EV-D68 VLP determined by ELISA. Error bars represent standard deviation (SD). In panels **b**, **c**, ZIKV-specific MAB 1C11 and HCV-specific MAB 1F4 served as IgG2a and IgG2b isotype controls (ctr), respectively.

Supplementary Figure 2. Reactivities of anti-EV-D68 MABs (2H12 and 8F12) towards EV71 VLP (**a**), and CVA16 VLP (**b**) determined by ELISA. Anti-EV71 MAB D5 and anti-CVA16 MAB 9B5 were used as positive controls for detection of EV71 VLP and CVA16 VLP, respectively. Error bars represent SD.

Q1-4: Just a suggestion, Fig. 7g could benefit from the use of the same arrow convention to indicate time of mAb administration that is used in Fig. 2a-c.

A1-4: The point is well taken. As suggested, we have modified Fig. 7g and have used the red arrow to indicate the time point of MAb injection.

(g, h) Therapeutic efficacy of chimeric MAbs against EV-D68 infection in neonatal mice. One-day-old ICR mice ($n = 14\text{--}15/\text{group}$) were inoculated i.p. with strain 18947. The suckling mice were i.p. injected with PBS, $10\ \mu\text{g/g}$ of c2H12, or $10\ \mu\text{g/g}$ of c8F12 at 1 dpi. Red arrow indicates time of MAb administration. Clinical scores were graded described in Figure 2 legend. Survival of mice in each antibody-treated groups was compared to the PBS control group. Statistical significance was indicated as follows: ***, $P < 0.001$. Error bars represent SEM.

Q1-5: I would appreciate if the authors could clarify their thoughts behind the design of the BLI competition assay depicted in Fig. 6e. I am accustomed to such assays having a few more experimental conditions, including the reciprocal blocking of saturating with 8F12 and then following with 2H12. Further, I am not accustomed to seeing the second antibody condition as a mixture. In other words, I would expect after saturating with 2H12 to then follow by dipping the sensors in a well containing only 8F12, not a combination of 2H12 and 8F12. Since extent of competition requires comparison to the level of binding of the 2nd mAb in the absence of the 1st mAb, binding of the 8F12 mAb alone should be shown alongside the 2H12 and 8F12 competition binding curve to allow the reader to interpret the extent of competition. Additionally, I have typically seen the binding curves corrected so that they normalize to the time of the addition of the second mAb, which is a function that the Octet software can perform. My comments aren't meant to invalidate this experiment, but to raise questions of why the experiment was designed as it was. Even though the conclusion that these mAbs have similar or largely overlapping epitopes is well made overall throughout the manuscript, certainly the reciprocal competition data would strengthen this conclusion and would have been easy to do alongside the included experiments, so is curious by its omission.

A1-5: Thanks for the comments. We used a combination of 2H12 and 8F12 as the second antibody condition in BLI competition assays, because we had observed that

both 2H2 and 8F12 had fast dissociation rate constants with K_{dis} being 2.2×10^{-3} and $3.5 \times 10^{-4} \text{ s}^{-1}$, respectively (Fig 1d-e, also shown below for easy view).

Fig 1 d, e Binding affinities of the MABs to EV-D68 18947 virion measured by BLI. Association and dissociation steps are divided by dotted red line. MAb concentrations used and values of KD, Kon and Kdis were shown.

Specifically, when MAb 2H12 was used as the first antibody, BLI signal significantly decreased during the dissociation phase (please see the “Buffer” line in the new Fig. 6e); if only 8F12 MAb alone was used as the second antibody, the 8F12 binding signal would inevitably be affected by the 2H12 dissociation signal, meaning that “the observed binding signal of 8F12 = the true binding signal of 8F12 - the reduced signal of 2H12” (this would make the data difficult to interpret especially when taking the association time of the second antibody into consideration). By contrast, when the mixture of 2H12 and 8F12 was used as the second antibody, 2H12 in the mixture could prevent the reduction of 1st antibody signal and, in this way, we could directly measure the true binding activity of the 8F12 antibody. The same is true when MAb 8F12 was used as the first antibody (new Fig. 6f). We feel that such a design (antibody mixture as the second antibody condition) is reasonable.

As suggested, we have also performed the reciprocal competition assay and the new data has been included in the revised manuscript (new Fig. 6f and also shown below for easy view). In addition, the related description/statements have been added, to read “To verify this structure-based prediction, we performed a BLI competitive binding assay, in which immobilized EV-D68 virion was saturated with the first antibody 2H12 and then allowed to bind the second antibody 8F12 in the presence of 2H12. As shown in the Fig. 6e, the initial binding of 2H12 to EV-D68 virions greatly blocked subsequent 8F12 binding. Similarly, when MAb 8F12 was used as the first antibody, the second antibody 2H12 produced very low BLI signal (Fig. 6f). These data suggest that 2H12 and 8F12 compete with each other for binding to EV-D68

virion”.

Fig 6 e, f Competitive binding assay using BLI. Immobilized EV-D68 virion (18947 strain) was saturated with the first antibody 2H12 (**e**) or 8F12 (**f**) and then allowed to react with the buffer, the antibody mixture (2H12 + 8F12), 2H12 alone (control) (**e**) or 8F12 alone (control) (**f**).

Q1-6: As another question of clarification, how did the authors determine which amino acids to label as the sialic acid binding site in orange in Fig. 4f and 5l? Did the authors just use the contact amino acids listed in Table S3 of reference 21? In looking at reference 21, visually the sialic acid binding site outlined in Fig. 2 and S2 of reference 21 has a different appearance than the sialic acid binding site labeled in Fig. 4f and 5l of this manuscript. Could the authors provide this clarification and an explanation if one is known about why these appear different?

A1-6: For the footprint generation, we downloaded the X-ray structure of EV-D68 in complex with the sialylated trisaccharide Neu5Ac α 2-6Gal β 1-4GlcNAc (EV-D68-6'SLN, PDB ID: 5BNO) from Ref 21 (note that the sialylated trisaccharide Neu5Ac α 2-6Gal β 1-4GlcNAc is preferentially recognized by EV-D68 according to Ref 21). We then superimposed the X-ray structure with our EV-D68 capsid model, and saved their sialic acid portion as a separate model relative to our capsid. Subsequently, we transformed the sialic acid model into a density map in XPLOR format that RIVEM could read. Finally, we used RIVEM to generate the footprints of sialic acid on EV-D68 shown in our manuscript. We did not use the contact amino acids information listed in Table S3 of Ref 21 to generate the footprints.

Regarding why the sialic acid binding site outlined in Fig. 2 and S2 of the Ref 21 has a different appearance than that shown in our figures (Fig. 4f and 5l), we should point out that the footprint for sialic acid in Ref 21 seemed to be summation of several sialic acids whereas our footprint was based on a single sialic acid model. Specifically, the

authors of Ref21 provided three EV-D68 in complex of sialic acid models (EV-D68-6'SL, EV-D68-3'SLN and EV-D68-6'SLN) in that paper, and described how they generated the footprints in Fig.2 legend as “The thick white contour outlines the summation of **five superimposed footprints** of Ig-like receptors on the virus, whereas the thinner white contour represents the consensus **footprint of at least four footprints. Shown also is the footprint of the sialylated trisaccharides (yellow).**” They referred to this figure in the main text as “**All three receptor analogues** were observed to bind near the ‘eastern end’ of the canyon (Fig. 2).” and “**In all three structures** the Neu5Ac moiety is ... (Fig. 2; Supplementary Fig. 2)”. Since they did not specify exactly which **sialylated trisaccharide structure** was used to generate Fig. 2 and S2, and based on their description it appears the footprints for sialic acid in these figures are a summation of all three structures. In addition, it's not clear whether they used density map or model as initials, which might further contribute to the difference. Since there are so many uncertainties, we chose to use a single model to generate sialic acid footprint instead. In short, our sialic acid footprint originate from one single model, whereas it is not clear whether the sialic acid footprint shown in Ref. 21 originated from density map or model and how many maps/models were used to generate the footprint --- all of these could contribute to the difference in appearance between the two footprints.

For clarification, we have now added details for sialic acid footprints generation in Figure legend and Method, to read, “Footprints of the sialic acid receptor (generated using the EV-D68-6'SLN model, PDB: 5BNO) are shown in orange” in legends for Fig. 4f and 5l, and “Roadmaps were generated by RIVEM (Radial Interpretation of Viral Electron density Maps)⁶². Footprint for sialic acid was generated using the EV-D68-6'SLN model (PDB: 5BNO)²¹.” in Method session.

REVIEWERS' COMMENTS

Reviewer #1 (Remarks to the Author):

Thank you for the thoughtful responses. My questions have been addressed.

Dear reviewer,

We would like to thank you for your feedback. We have responded to the points raised by the reviewer and highlighted the changes in yellow in the revised text.

Response to reviewer #1' s comments:

Reviewer #1 (Remarks to the Author):

Thank you for the thoughtful responses. My questions have been addressed.

Response: Thanks.

Second round of review

Response to reviewer #1's comments:

Reviewer #1 (Remarks to the Author):

The authors have returned a revised manuscript that is greatly strengthened by additional clarifications of their structural studies, rephrasing of key concepts around chimerism vs. humanization and synergy vs. additive effects, and addition of suggested experiments. I have the following minor comments:

Response: We thank the reviewer for the positive comments. Below are our responses to the reviewer's specific concerns.

Q1-1: In line 431 the authors still use the word "humanized." They should either clarify that the experiments in this manuscript are intended to give credence to the idea that humanization is a worthy effort, or they should drop the use of the term humanized. Because as stated, this implies that the experiments in this paper will determine whether humanized MAbs are efficacious, and that is not the case.

A1-1: Thanks for the comments. As suggested, the word "humanized" in line 431 has been replaced with the word "chimerized" in our revised manuscript, to read "we then asked whether these two murine MAbs could be chimerized with human IgG constant regions".

Q1-2: In the methods the authors still need to specify that the antibodies used in the western blots are constant domain-specific. They provide a level of detail in their response letter in the beginning portion of A3-8 that will help a general reader and could essentially be copied into the methods.

A1-2: Thanks for the suggestions. We have modified the Methods section to include this description, to read "Western blotting of chimeric MAbs was carried out as described previously ⁶³, except that HRP-conjugated goat anti-mouse IgG (Sigma, USA) and HRP-conjugated rabbit anti-human IgG (Abcam, USA) were used for detection. Note that anti-human IgG or anti-mouse IgG secondary antibodies (HRP conjugate) used in this study recognize the constant domains of both heavy and light chains." (please see lines 796-798).

Q1-3: The labeling of the x-axis for ELISA data in Fig. 1c. and S2 is atypical. The authors show total antibody amounts (ng/well) rather than concentration of antibody (such as ng/mL or µg/mL) when displaying ELISA binding data. It would be more typical to label with antibody concentrations, and this would be easy enough to convert. This will also make presentation of the ELISA data the same as the neutralization and HAI data, in which concentration is used rather than amount of

antibody.

A1-3: The point is well taken. As suggested, we have modified the labeling of the x-axis for ELISA data in Fig. 1c. and S2 and have shown the antibody concentration instead of antibody amount. For the convenience of the reviewer, we also show the figures here.

Fig. 1 c Reactivities of the MAbs towards EV-D68 VLP determined by ELISA. Error bars represent standard deviation (SD). In panels **b**, **c**, ZIKV-specific MAb 1C11 and HCV-specific MAb 1F4 served as IgG2a and IgG2b isotype controls (ctr), respectively.

Supplementary Figure 2. Reactivities of anti-EV-D68 MAbs (2H12 and 8F12) towards EV71 VLP (**a**), and CVA16 VLP (**b**) determined by ELISA. Anti-EV71 MAb D5 and anti-CVA16 MAb 9B5 were used as positive controls for detection of EV71 VLP and CVA16 VLP, respectively. Error bars represent SD.

Q1-4: Just a suggestion, Fig. 7g could benefit from the use of the same arrow convention to indicate time of mAb administration that is used in Fig. 2a-c.

A1-4: The point is well taken. As suggested, we have modified Fig. 7g and have used the red arrow to indicate the time point of MAb injection.

(g, h) Therapeutic efficacy of chimeric MABs against EV-D68 infection in neonatal mice. One-day-old ICR mice ($n = 14\text{--}15/\text{group}$) were inoculated i.p. with strain 18947. The suckling mice were i.p. injected with PBS, $10\ \mu\text{g/g}$ of c2H12, or $10\ \mu\text{g/g}$ of c8F12 at 1 dpi. Red arrow indicates time of MAb administration. Clinical scores were graded described in Figure 2 legend. Survival of mice in each antibody-treated groups was compared to the PBS control group. Statistical significance was indicated as follows: ***, $P < 0.001$. Error bars represent SEM.

Q1-5: I would appreciate if the authors could clarify their thoughts behind the design of the BLI competition assay depicted in Fig. 6e. I am accustomed to such assays having a few more experimental conditions, including the reciprocal blocking of saturating with 8F12 and then following with 2H12. Further, I am not accustomed to seeing the second antibody condition as a mixture. In other words, I would expect after saturating with 2H12 to then follow by dipping the sensors in a well containing only 8F12, not a combination of 2H12 and 8F12. Since extent of competition requires comparison to the level of binding of the 2nd mAb in the absence of the 1st mAb, binding of the 8F12 mAb alone should be shown alongside the 2H12 and 8F12 competition binding curve to allow the reader to interpret the extent of competition. Additionally, I have typically seen the binding curves corrected so that they normalize to the time of the addition of the second mAb, which is a function that the Octet software can perform. My comments aren't meant to invalidate this experiment, but to raise questions of why the experiment was designed as it was. Even though the conclusion that these mAbs have similar or largely overlapping epitopes is well made overall throughout the manuscript, certainly the reciprocal competition data would strengthen this conclusion and would have been easy to do alongside the included experiments, so is curious by its omission.

A1-5: Thanks for the comments. We used a combination of 2H12 and 8F12 as the second antibody condition in BLI competition assays, because we had observed that both 2H2 and 8F12 had fast dissociation rate constants with K_{dis} being 2.2×10^{-3} and $3.5 \times 10^{-4}\ \text{s}^{-1}$, respectively (Fig 1d-e, also shown below for easy view).

Fig 1 d, e Binding affinities of the MAbs to EV-D68 18947 virion measured by BLI. Association and dissociation steps are divided by dotted red line. MAb concentrations used and values of KD, Kon and Kdis were shown.

Specifically, when MAb 2H12 was used as the first antibody, BLI signal significantly decreased during the dissociation phase (please see the “Buffer” line in the new Fig. 6e); if only 8F12 MAb alone was used as the second antibody, the 8F12 binding signal would inevitably be affected by the 2H12 dissociation signal, meaning that “the observed binding signal of 8F12 = the true binding signal of 8F12 - the reduced signal of 2H12” (this would make the data difficult to interpret especially when taking the association time of the second antibody into consideration). By contrast, when the mixture of 2H12 and 8F12 was used as the second antibody, 2H12 in the mixture could prevent the reduction of 1st antibody signal and, in this way, we could directly measure the true binding activity of the 8F12 antibody. The same is true when MAb 8F12 was used as the first antibody (new Fig. 6f). We feel that such a design (antibody mixture as the second antibody condition) is reasonable.

As suggested, we have also performed the reciprocal competition assay and the new data has been included in the revised manuscript (new Fig. 6f and also shown below for easy view). In addition, the related description/statements have been added, to read “To verify this structure-based prediction, we performed a BLI competitive binding assay, in which immobilized EV-D68 virion was saturated with the first antibody 2H12 and then allowed to bind the second antibody 8F12 in the presence of 2H12. As shown in the Fig. 6e, the initial binding of 2H12 to EV-D68 virions greatly blocked subsequent 8F12 binding. Similarly, when MAb 8F12 was used as the first antibody, the second antibody 2H12 produced very low BLI signal (Fig. 6f). These data suggest that 2H12 and 8F12 compete with each other for binding to EV-D68 virion”.

Fig 6 e, f Competitive binding assay using BLI. Immobilized EV-D68 virion (18947 strain) was saturated with the first antibody 2H12 (**e**) or 8F12 (**f**) and then allowed to react with the buffer, the antibody mixture (2H12 + 8F12), 2H12 alone (control) (**e**) or 8F12 alone (control) (**f**).

Q1-6: As another question of clarification, how did the authors determine which amino acids to label as the sialic acid binding site in orange in Fig. 4f and 5l? Did the authors just use the contact amino acids listed in Table S3 of reference 21? In looking at reference 21, visually the sialic acid binding site outlined in Fig. 2 and S2 of reference 21 has a different appearance than the sialic acid binding site labeled in Fig. 4f and 5l of this manuscript. Could the authors provide this clarification and an explanation if one is known about why these appear different?

A1-6: For the footprint generation, we downloaded the X-ray structure of EV-D68 in complex with the sialylated trisaccharide Neu5Ac α 2-6Gal β 1-4GlcNAc (EV-D68-6'SLN, PDB ID: 5BNO) from Ref 21 (note that the sialylated trisaccharide Neu5Ac α 2-6Gal β 1-4GlcNAc is preferentially recognized by EV-D68 according to Ref 21). We then superimposed the X-ray structure with our EV-D68 capsid model, and saved their sialic acid portion as a separate model relative to our capsid. Subsequently, we transformed the sialic acid model into a density map in XPLOR format that RIVEM could read. Finally, we used RIVEM to generate the footprints of sialic acid on EV-D68 shown in our manuscript. We did not use the contact amino acids information listed in Table S3 of Ref 21 to generate the footprints.

Regarding why the sialic acid binding site outlined in Fig. 2 and S2 of the Ref 21 has a different appearance than that shown in our figures (Fig. 4f and 5l), we should point out that the footprint for sialic acid in Ref 21 seemed to be summation of several sialic acids whereas our footprint was based on a single sialic acid model. Specifically, the authors of Ref21 provided three EV-D68 in complex of sialic acid models

(EV-D68-6'SL, EV-D68-3'SLN and EV-D68-6'SLN) in that paper, and described how they generated the footprints in Fig.2 legend as “The thick white contour outlines the summation of **five superimposed footprints** of Ig-like receptors on the virus, whereas the thinner white contour represents the consensus **footprint of at least four footprints**. **Shown also is the footprint of the sialylated trisaccharides (yellow)**.” They referred to this figure in the main text as “**All three receptor analogues** were observed to bind near the ‘eastern end’ of the canyon (Fig. 2).” and “**In all three structures** the Neu5Ac moiety is ... (Fig. 2; Supplementary Fig. 2)”. Since they did not specify exactly which **sialylated trisaccharide structure** was used to generate Fig. 2 and S2, and based on their description it appears the footprints for sialic acid in these figures are a summation of all three structures. In addition, it's not clear whether they used density map or model as initials, which might further contribute to the difference. Since there are so many uncertainties, we chose to use a single model to generate sialic acid footprint instead. In short, our sialic acid footprint originate from one single model, whereas it is not clear whether the sialic acid footprint shown in Ref. 21 originated from density map or model and how many maps/models were used to generate the footprint --- all of these could contribute to the difference in appearance between the two footprints.

For clarification, we have now added details for sialic acid footprints generation in Figure legend and Method, to read, “Footprints of the sialic acid receptor (generated using the EV-D68-6'SLN model, PDB: 5BNO) are shown in orange” in legends for Fig. 4f and 5l, and “Roadmaps were generated by RIVEM (Radial Interpretation of Viral Electron density Maps)⁶². Footprint for sialic acid was generated using the EV-D68-6'SLN model (PDB: 5BNO)²¹.” in Method session.

First round of review

Response to reviewer #1's comments:

Reviewer #1 (Remarks to the Author):

In this manuscript Zhang, et al detail a study of two murine monoclonal antibodies (mAbs) isolated by the authors after immunization of mice with enterovirus D68 (EV-D68) virus-like particles. These two mAbs have unique neutralization profiles across different clades of EV-D68 despite binding to nearly identical epitopes as identified by the authors using cryo-EM. Further, they performed mechanistic studies to identify how the antibodies neutralize EV-D68, by blocking interactions with sialic acid and inhibiting attachment to cells in vitro. The cryo EM structures also indicate that one of the mAbs, 2H12, may additionally neutralize by capturing a novel structural intermediate of the virus that can allow escape of the genomic material. Finally, they show that both native murine and chimeric human forms of the mAbs can protect mice from EV-D68 challenge when given up to 3 or 1 days after infection, respectively.

The epitopes identified are unique from the only other structurally identified EV-D68 mAb epitopes in the literature: two murine mAbs in reference 19 and two human mAbs in reference 25. The macaque mAb epitope in reference 24 was not confirmed with structural methods. This represents an advance in the understanding of how antibodies can interact with EV-D68, a medically significant virus responsible for causing outbreaks of a severe paralyzing illness, acute flaccid myelitis. The difference in neutralization mechanisms despite nearly overlapping epitopes is also of general interest in the understanding of antibody-virus interactions.

The work in this paper is quite thorough and generally well arranged and easy to follow.

Response: We thank the reviewer for the positive comments. Below are our responses to the reviewer's specific concerns.

Q1-1: There are a number of statements of primacy, though, that are unfounded or unnecessary. This is especially noted with the use of "elite" in line 408 and throughout lines 72-83. For example, the study in reference 25 does in fact test the therapeutic window of giving mAbs at different time points after infection in mice, which is counter to the claims in lines 78-79. Based on this, the use of "unprecedented" in line 455 is not applicable.

A1-1: Thanks for the comments. As suggested, we have toned down the statements and made modifications accordingly in our revised manuscript. Specially, we **deleted** "However, in the three studies^{19, 24, 25}, therapeutic effectiveness against diverse EV-D68 clades and/or therapeutic window of the anti-EV-D68 MAbs have not been demonstrated, making these MAbs less attractive for further development into

therapeutics for future application in real-life clinical settings. Therefore, it is important to continue searching for more powerful anti-EV-D68 MAbs suitable for developing MAb-based therapy for treating EV-D68 infections in human” (previous lines 77-83). In addition, the description “an elite drug candidate” (previous line 408) was modified as “a strong drug candidate”. The statement “unprecedented therapeutic potency and breadth” (previous line 455) was modified as “remarkable therapeutic potency and breadth”.

Q1-2: It is also curious that the discussion in lines 410-431 compares the mAbs in this study to prior referenced murine and macaque mAbs but not the human mAbs, which have equivalent to stronger neutralization potency against the same EV-D68 isolates. In any case, it is difficult to quantitatively compare neutralization potencies across different studies because they are measured using different techniques in different laboratories. The studies in this paper are of sufficient interest and quality that the comparisons are not necessary to bolster the significance.

A1-2: Thanks. We have followed the suggestion to delete the statement about the comparison of neutralization capacity of different antibodies (previous lines 410-431).

Q1-3: The statement in lines 220-222 could be reworded to improve accuracy, as the mAb cocktail was never superior to a single mAb in any of the in vivo experiments. This statement and the use of the word “synergy” in line 159 seem to imply that the antibody cocktail of 2H12 and 8F12 is superior in efficacy to either mAb individually. That would demonstrate synergy. What the authors show is simply that each antibody individually is more effective at neutralizing or protecting against some EV-D68 strains than others. Therefore, giving a cocktail of mAbs provides redundancy, but there is no synergy. The combination of mAbs is no more effective than the most effective individual mAb in the cocktail.

A1-3: We agree with your opinion. We apologize for the unclear descriptions. To avoid confusion, we have modified the manuscript accordingly. The statement “the 2H12/8F12 cocktail is superior to single MAbs in treating diverse EV-D68 infections at a delayed time point” (previous lines 220-222) was modified as follows: “the above results demonstrate that the 2H12/8F12 cocktail is very effective in treating diverse EV-D68 infections at a delayed time point”. In addition, the description “to create synergy” (previous line 159) was deleted.

Q1-4: The authors use the term “humanized” (first used in line 374) in a manner inconsistent with the antibody field. The antibodies used in this study are mouse-human chimeras, or “chimeric” antibodies. The term humanized more specifically implies that only the CDR loops of the murine mAb were used in place of

the CDRs of a human mAb backbone. The authors swapped out the entire variable regions, though, which is not truly “humanization.” This has important clinical implications because truly humanized mAbs are less immunogenic to humans than the mouse-human chimeras described in this manuscript. Variations of the word humanized should be eliminated from the manuscript and replaced with variations of the word chimeric.

-Description of chimeric antibodies: <https://absoluteantibody.com/our-technology/formats-we-have-made/chimeric-antibodies/>

-Description of how palivizumab was humanized, swapping out only the CDRs: S Johnson et al, J Infect Dis, 1998; PMID: 9359721 DOI: 10.1086/514115

A1-4: Thanks for pointing this out. As suggested, the word “humanized” has been replaced with the words “human-mouse chimeric” or “chimeric” in our revised manuscript.

Q1-5: The authors should further clarify that the specific antibodies they use for the western blot in Fig. 7b, discussed in lines 380-382, are Fc-specific. This is not mentioned in the methods or in the main text. One would otherwise expect a polyclonal anti-mouse IgG antibody to react to both the native murine mAb and the human-chimeric mAb.

A1-5: Thanks for the suggestion. We have modified the manuscript accordingly (please see lines 437-438), to read “Chimeric MAbs 2H12 (c2H12) and 8F12 (c8F12) reacted with anti-human IgG (**constant domains**) antibody but not anti-mouse IgG (**constant domains**) antibody in western blot assays”. In addition, the statement “Anti-human IgG or anti-mouse IgG secondary antibodies (HRP conjugate) used in this study recognize the constant domains of both heavy and light chains” has been included in Fig. 7b legend.

Q1-6: The authors frequently refer to reference 32 (lines 401 and 432) in describing a 1-3 day incubation period for EV-D68 in humans. This reference is a small study that specifically refers to respiratory illness. The therapeutic mouse models described in this manuscript focus on a neurologic phenotype (paralysis) related to EV-D68 infection. Much larger studies have described median 5 days of illness preceding limb weakness in AFM, the typical human neurologic phenotype of EV-D68 infection. This longer period of incubation to AFM should be acknowledged in the manuscript.

-Study showing 5 day incubation: J Lopez et al, MMWR 2019, PMID: 31295232 DOI: 10.15585/mmwr.mm6827e1

A1-6: Thanks for the constructive comments. As suggested, we cited the paper “Vital Signs Surveillance for Acute Flaccid Myelitis — United States, 2018” and modified

the statements (lines 458-460), to read “a prodromal illness (such as febrile, respiratory, and/or gastrointestinal) preceded the onset of limb weakness in most AFM patients by a median of 5 days^{32, 33}” (Ref 32. Lopez A, et al. Vital Signs: Surveillance for Acute Flaccid Myelitis - United States, 2018. MMWR Morb Mortal Wkly Rep 68, 608-614 (2019); ref 33. Messacar K, et al. Acute flaccid myelitis: A clinical review of US cases 2012-2015. Ann Neurol 80, 326-338 (2016)).

Q1-7: There are a few graphs in the paper that appear to be lacking error bars, specifically the ELISA in Fig. S2 and the clinical scores in Figs. 2, S3B, and 7H. Regarding the clinical scores, it is impossible to understand the heterogeneity in clinical disease in the mice without them. Alternatively, the authors could create a supplemental table listing the clinical scores for each mouse to give an idea of this heterogeneity.

A1-7: Thanks for the constructive comments. There are error bars in Fig. S2, but they are too small to be visible. In addition, as suggested, we have remade the graphs (Figs. 2, S3B, and 7H) with error bars, by entering raw data (clinical scores of each mouse) instead of averaged data in the GraphPad Prism software.

Fig. 2 Therapeutic efficacy of the MAbs against EV-D68 infection in mice. One-day-old ICR mice (n = 13–27/group) were inoculated intraperitoneally (i.p.) with strain 18947 **a-b** or strain

18953 **c**. The suckling mice were i.p. injected with PBS, 10 µg/g of 2H12, 10 µg/g of 8F12, or a mixture of both MAbs (10 µg/g of each MAb) at 1 day post-infection (dpi) **a** or 3 dpi **b**, **c** and were then monitored daily for survival and clinical score. Clinical scores were graded as follows: 0, healthy; 1, lethargy and reduced mobility; 2, limb weakness; 3, limb paralysis; 4, death. Survival rates of antibody-treated mice were compared with the mice in the PBS control group. Statistical significance was indicated as follows: ns., no significant difference ($P \geq 0.05$); *, $P < 0.05$; ***, $P < 0.001$. Note that to prevent overlap, the overlapping data sets in left panels (survival curves) were nudged by 5 units in the Y direction. All error bars represent SEM.

Fig. 7. (g, h) Therapeutic efficacy of chimeric MAbs against EV-D68 infection in neonatal mice. One-day-old ICR mice ($n = 14\text{--}15/\text{group}$) were inoculated i.p. with strain 18947. The suckling mice were i.p. injected with PBS, 10 µg/g of c2H12, or 10 µg/g of c8F12 at 1 dpi. Clinical scores were graded described in Figure 2 legend. Survival of mice in each antibody-treated groups was compared to the PBS control group. Statistical significance was indicated as follows: ***, $P < 0.001$. Error bars represent SEM.

Supplementary Figure 3. Prophylactic efficacy of MAbs 2H12 and 8F12 against EV-D68 infection in neonatal mice. Groups of ICR mice (age $< 24\text{h}$; $n = 12\text{--}14/\text{group}$) were i.p. injected with PBS, 10 µg/g of 2H12, 8F12, IgG2a isotype control (ctr) MAb (1C11), or IgG2b isotype control MAb (1F4) and one day later challenged with strain 18947. The challenged mice were monitored daily for (a) survival and (b) clinical score. Clinical scores were graded as follows: 0, healthy; 1, lethargy and reduced mobility; 2, limb weakness; 3, limb paralysis; 4, death. Survival of mice in each antibody-treated groups was compared to the PBS control group. Statistical significance was indicated as follows: ns., no significant difference ($P \geq 0.05$);

** , P < 0.01; ***, P < 0.001. Error bars represent SEM.

Response to reviewer #2' s comments:

Reviewer #2 (Comments to the Author):

This is an important work to characterize EV-D68 neutralizing Mabs, 2H12 and 8F12. Their efficacy as a cocktail for treatment of infections is explored by multiple assays including a structural characterization. The largest concern is a lack of careful quantification to accompany the structural analysis. There are a few other minor concerns, necessary clarifications, and areas where more information is needed.

Response: We thank the reviewer for the positive comments on our work.

Q2-1: Line 93 and elsewhere, 2.9 is not atomic resolution and should be described as high resolution or near atomic resolution.

A2-1: The point is well taken. We have modified related descriptions accordingly in our revised manuscript (please see lines 91, 273, and 504), to read “The 2.9-Å-resolution cryo-electron microscopy (cryo-EM)”, “illustrating the high resolution of the map”, and “Importantly, our high-resolution cryo-EM structures revealed”.

Q2-2: Line 142 Bio-layer interferometry (BLI) is not defined or cited when first used in the Results.

A2-2: Thanks for pointing this out. As suggested, we have defined this term in our revised manuscript, to read “determine the binding affinity of the MAbs to the 18947 virion by bio-layer interferometry (BLI)” (please see line 140).

Q2-3: Manuscript would greatly benefit from a table listing the different strains and each source and abbreviation used throughout.

A2-3: Thanks for the constructive suggestion from the reviewer. We have now added a table to summarize the information of EV-D68 strains used in our study as Supplementary Table 1 in our reviewed manuscript. For the convenience of the reviewers and editor, we also show it here.

Supplementary Table 1. A summary of all the EV-D68 strains used in this study.

EV-D68 strains	Abbreviation	Clade	Source (catalog)	Genbank ID
US/MO/14-18947	18947	B	ATCC (VR-1823)	KM851225
US/MO/14-18950 ^a	18950	B	P1 gene was synthesized	KM851228
US/KY/14-18953	18953	D	ATCC (VR-1825)	KM851231
Fermon	Fermon	Prototype	ATCC (VR-1826)	AY426531

^a VLP vaccine strain.

Q2-4: There is not enough description for the cryo EM section. Description of the unchanged virus capsid should include quantification. What is the RMSD of the capsid superimposed with virus versus A-particle structures? Such a comparison should be the basis of a conclusion that the FAB does not induce change, not a visual inspection and speculation that the two fold is closed or open.

A2-4: The point is well taken. We have calculated the RMSD values of the capsid proteins between our structures and the superimposed full native or A-particle of the EV-D68 structure, which is now provided as Supplementary Table 3 in our revised manuscript. The RMSDs are calculated based on the maximum shared C α atoms of protomers in each pair of molecules excluding the VP4 subunit. We have also modified the text accordingly, to read “The capsid structure in the EV-D68/8F12 complex is essentially identical to that of the unbound native EV-D68 virion¹⁸ with the overall C α root-mean-square deviation (RMSD) value being rather small as 0.53 Å between the protomers of the two structures (Supplementary Table 3); also, the two-fold axis channel remains closed in the complex (Fig. 4c). Collectively, these results indicate that 8F12 Fab binding does not induce obvious conformational changes of the EV-D68 virion”, and “We compared our structures with the available cryo-EM structures of the virion (PDB: 6CSG), expanded 1 (E1, PDB: 6CS3), and A particle (PDB: 6CS6) of the 18947 strain. As shown in Supplementary Table 3, the overall C α RMSD between the capsid protomer of the 18947 virion and that of our S1 state was calculated to be as small as 0.86 Å, indicating that the viral particle in S1 adopts the native conformation. A better fit in capsid protomer structure was observed between our S2 state and the 18947 E1-particle (RMSD = 1.21 Å) than between our S2 state and the 18947 native virion (RMSD: 1.99 Å) or A-particle (RMSD: 3.52 Å), suggesting that S2 is more close to E1 conformation and does not resemble A-particle. Our S3 state fits better with the 18947 A-particle in the capsid protomer (RMSD: 1.61 Å) than the other two states, but with significantly weaker RNA genome density than typical A-particle (Fig. 5i).” (please see lines 282-286 and 350-361).

Supplementary Table 3. RMSD values between capsid protomers.

RMSD (Å)	EV-D68/8F12	EV-D68/2H12 (S1)	EV-D68/2H12 (S2)	EV-D68/2H12 (S3)
Full native EV-D68 (PDB: 6CSG)	0.53	0.86	1.99	2.51
A-particle EV-D68 (PDB: 6CS6)	4.09	4.20	3.52	1.61
E1-particle EV-D68 (PDB: 6CS3)			1.21	1.86

Q2-5: Line 316, in all reports to data, genome release at the two fold is suggested.

This is a model not proven yet, but merely suggested by numerous A-particle structures.

A2-5: Thanks for the suggestion. We have modified the related statement, to read “Noteworthy, enterovirus genomes have been **suggested** to be released through the two-fold axis channel^{30, 31}”.

Q2-6: Include a central section for 8F12 complex map as well so that the quality of the density can be seen.

A2-6: The suggestion is well taken. We have now included this central section of EV-D68/8F12 complex map in the revised Supplementary Fig. 4e (we also show it below for the convenience of the reviewer and editor) and added related description in manuscript, to read “Central section of the EV-D68/8F12 complex map is provided in Supplementary Fig. 4e, to show the quality of the densities corresponding to the Fab, viral capsid and RNA.” in lines 274-276.

Supplement Figure 4e. Central section for EV-D68/8F12 complex map.

Q2-7: Building the variable domains of the antibody models should be described prior to reporting on the orientation of the light and heavy chains relative to the capsid.

A2-7: We followed the suggestion from our reviewer to make the description for model building of Fab more clear by adding the following sentence “Note that only models of the variable regions of the Fab were built, because electron densities corresponding to the constant regions were relatively weak (more details in the Methods).” (please see lines 276-278, lines 319-320).

Q2-8: RIVEM should also be cited in the fig 4 legend where mentioned.

A2-8: The suggestion is well taken, and we have now added the citation for RIVEM in Fig.4 legend, to read “This figure was generated using RIVEM (Radial Interpretation of Viral Electron density Maps; reference [62])” .

Q2-9: How was buried surface determined?

A2-9: We used the PISA server (https://www.ebi.ac.uk/pdbe/prot_int/pistart.html)¹ to determine the buried surface, which is a commonly used software for this purpose².

We first uploaded the coordinate file containing a protomer and the associated Fab, then the PISA server could calculate the interface area between chains, which is an average of solvent-accessible area in interface from the two calculated chains. We have modified the related methods and Supplementary Table 6 in the revised manuscript, to read “Fab-virion interaction analysis including hydrogen bond, salt bridge prediction, **and buried surface area** were carried out through PISA server (https://www.ebi.ac.uk/pdbe/prot_int/pistart.html)”, and “Surface area buried by V_H and V_L of MAb 8F12 and 2H12 in state S1 **determined by the PISA server** and their contribution to the total interface area” (please see lines 776-778 and Supplementary Table 6).

Q2-10: Please include more detail about how three conformational states induced by 2H12 were derived. How did the 2D classes correlate? How was radius determined?

A2-10: The suggestion is well taken. We have now added more details on how the three conformational states induced by 2H12 were derived in the updated Methods section, to read “Similar processing tactics were applied to EV-D68/2H12 dataset (Supplementary Fig. 6). After 2D classification, 8,844 particles remained. After CTF refinement and Bayesian polishing, these particles were reconstructed into a map at 3.25 Å resolution. Further 3D classification of this dataset into three classes revealed two distinct major conformations with better structural features (class 1 and 3, Supplementary Fig. 6a). Class 1 map depicts the full native virion feature and no other conformation could be detected after further 3D classification, which was thus termed S1 state. Meanwhile, class 3 map exhibits open channels at the two-fold axis, a characteristic feature of expanded enteroviral particles. We then performed further 3D classification on this dataset and obtained two distinct conformational states (termed S2 and S3, Supplementary Fig. 6a). Compared with S2, S3 shows different internal genomic RNA organization and significantly weaker RNA genome density. Although the overall capsid structures of S2 and S3 resemble each other, there are still some differences, for example, density for VP4 is totally missing in S3, which is not the case in S2.” (please see lines 751-759).

For the convenience of our reviewers and editor, we also show Supplementary Fig. 6a below.)

Supplementary Figure 6. Cryo-EM structural analysis of EV-D68/2H12 complex. (a) Representative cryo-EM image of the EV-D68/2H12 complex and the flowchart of the 3D reconstruction process for the EV-D68/2H12 complex. Full particle, partially solid-core particle, and empty particles are marked in red, yellow, and green boxes, respectively. Yellow arrow indicates the broken particle induced by 2H12 binding and the exposed viral genome.

We computationally sorted out these three states in the EV-D68/2H12 dataset, and some of the subtle feature differences such as the open channels at the two-fold axis is hard to be directly visualized in the reference-free 2D class averages. Still, for the overall genome occupancy status, the 2D analysis displays different characteristics, for instance, S1 showed representative characteristics of full native virion with strong RNA content, the RNA content density of S2 is comparable to S1, while for S3, the internal genome density was significantly weaker than that of S1 and S2 (Fig. R1). These 2D average characteristics are consistent with the observations in the related 3D reconstructions.

Fig. R1 Representative reference-free 2D class averages of (a) S1, (b) S2, and (c) S3 states for the EV-D68/2H12 complex. Density for inner RNA genome is weaker in S3 than the other two states.

For the radius measurement in Fig. S8, we first measured the diameter of the virus capsid through measuring the distance between the CG2 atom in residue V231 of VP1 from protomers located in the opposite direction on the diagonal, then divided the diameter by two to deduce the radius value. This atom is located at the most protruding part of the virus capsid. We have included this information in the legend of Supplementary Fig. 8.

Q2-11: A better structural comparison of S2 and S3 is needed (other than the supplemental figure 8 that is not very informative). What is the RMSD between S2 and S3? If these are in fact different conformations, we need to see the superimposition of the structural proteins and a description of the similarities and differences. Further issues with the interpretation that S2 and S3 are different conformations. The difference in diameter of 1 Angstrom between S2 and S3 is not significant (not believable) at this resolution.

A2-11: Thanks for the constructive suggestion from the reviewer. We have now provided more detailed structural comparison between S2 and S3 states (new Supplementary Fig. 8b-8e), to read “Furthermore, we made more detailed structural comparison between the S2 and S3 states by superimposition of the S2 and S3 protomer structures (Supplementary Fig. 8b-8e). The overall C α RMSD value between the two structures was calculated to be 1.47 Å, indicating conformational variations between the two states. For instance, the N-terminal of VP1 is externalized

and the first 40 residues are missing in S3 while most of them could be resolved and lie underneath VP3 in S2; the VP1 C-terminal loop (residues K270 to T289) is largely missing in S3 while captured in S2 (Supplementary Fig. 8c). Noteworthy, VP4 is completely missing in S3, while part of VP4 is present in S2 structure. Moreover, residues P43 to T54 from VP2 AB loop are also missing in S3 but captured in S2 (Supplementary Fig. 8d); residues P168 to T174 within VP3 GH loop display significantly different conformations in the two states (Supplementary Fig. 8e). Collectively, S2 and S3 are indeed different states in capsid conformation” in lines 337-349.

Supplementary Figure 8. Structural comparison of EV-D68/8F12 and EV-D68/2H12 (states S1, S2, and S3) structures. (b-e) Structural comparison of the protomers (b) and individual subunit proteins (c-e) of the EV-D68/2H12 S2 (color) and S3 (gray) states. The major conformational differences between the two states are indicated by arrows.

We agree with this reviewer that the difference in diameter of 1 Å between S2 and S3 is not significant. We just listed the number we measured to address that both S2 and S3 were expanded compared to S1 and didn't imply there was any notable difference in diameter between S2 and S3. We should emphasize that the main differences between S2 and S3 lie in the capsid protein composition, conformation, and RNA density as mentioned above.

Q2-12: For accurate genome comparison, the capsid density should be subtracted for

a 2D classification. One cannot rely on 3D classification to compare genome density as a few empty particles could be included in the potential A-particle class.

A2-12: Thanks for the suggestion from our reviewer. We agree that 3D classification may be not enough and 2D classification on subtracted particles is a good approach. As suggested, we subtracted the capsid density of virion particles in the EV-D68/2H12 S2 and S3 datasets, and performed reference-free 2D classification with a mask of 300 Å indiameter to further exclude the influence of capsid. All the 2D class averages have inner content and no obviously empty particle classes were found as concerned. The resulting capsid-subtracted 2D class averages of S2 and S3 datasets show different inner genome organization (Fig. R2). For instance, the inner genomic RNA appears full and tightly packed in S2, while that in S3 appears sparse and less dense. These characteristics indicate that S3 possesses less genome content than S2. This conclusion is consistent with our 3D classification results.

Fig. R2 Reference-free 2D class averages of capsid-subtracted particles for EV-D68/2H12 S2 (a) and S3 (b) datasets.

Q2-13: How well does S2 align to A-particle (135S) structure?

A2-13: We should point out that S2 resembles the expanded 1 (E1) state, but not A particle, as stated in our initial manuscript and further substantiated by RMSD analysis (Supplementary Table 3). Specially, C α RMSD between S2 and A-particle (PDB: 6CS6) protomer is 3.52 Å, while C α RMSD between S2 and E1 (PDB: 6CS3) is reduced to 1.21 Å.

Still, we followed the suggestion from the reviewer to align S2 with A-particle (Fig.

R3), which reveals that there are multiple conformational variances between the two states. In state S2, the N-terminus of VP1 and VP4 is ordered and located inside the capsid shell; whereas in A particle, the N-terminal tail of VP1 is externalized and VP4 is missing (expelled from the capsid)³. These main characteristics distinguish S2 from A-particle. Besides, there are other obvious differences. For instance, the C-terminal residues K270 to T289 of VP1 are missing in A-particle but resolved in S2; there is a shift in VP1 β -barrel; VP2 AB loop, VP2 C-terminus, and VP3 GH-loop also show significant conformational differences between S2 and A-particle (Fig. R3). Collectively, these data demonstrate that viral capsid of S2 is different from that of A-particle.

Fig. R3 Structural comparison of the protomers (a) and individual subunit proteins (b-d) of the EV-D68/2H12 S2 (color) and A-particle (gray) states. The major structural differences between the two structures are indicated by arrows.

Q2-14: It cannot be understood from the micrograph shown in supplemental if there are any empty particles in the data. Perhaps the S2 and S3 classes are mixtures of A-particle and empty particles. Possibly the class corresponding to S2 is mostly comprised of A-particles and the S3 has more 80S empty particles.

A2-14: As shown in our raw micrographs (Fig. R4a), the majority of the particles in our sample are particles with genomes, and there are seldom empty particles. We only chose genome RNA-containing particles for subsequent 3D reconstruction. Moreover, we run multiple rounds of 2D classification to exclude the empty particles. 2D

classification of the remaining 8,844 particles used for final structural determination also confirmed that there is no empty particles in the dataset (Fig. R4b). Therefore, the particles used for 3D construction are all RNA-filled particles. Therefore, S3 (new intermediate with weak RNA density) is different from 80S empty particles (no RNA).

As mentioned above, viral capsid in S2 is different from that in A-particle (please see A2-13).

Fig. R4. Inspection of the EV-D68/2H12 dataset. **(a)** Raw Cryo-EM micrographs of the EV-D68/2H12 complex. Bar = 50 nm. Full particle, partially solid-core particle, and empty particles are marked in red, yellow, and green boxes, respectively. Yellow arrow indicates the broken particle induced by 2H12 binding and the exposed viral genome. **(b)** 2D class averages of the remaining 8,844 particles used for final structural determination for EV-D68/2H12 complex.

Q2-15: In Fig 5 a, b, and c (S1, 2, and 3) are reported to be colored radially according to the same key, but S2 has more expansion (more red showing) than S3. Was the color key altered for C?

A2-15: The color key remains the same for Fig. 5a, b, and c. Here the red color represents smaller radius within 100 Å, corresponding to the enclosed interior genome density, instead of illustrating expansion. The S2 shows more inner red density than S3 is because though S2 and S3 has the same level of expansion, S2 has more genomic RNA density in its inner space than that of S3 (Fig. 5 h-i, and Fig. R2).

Response to reviewer #3' s comments:

Reviewer #3 (Comments to the Author):

Review Summary:

The study reports two new EV-D68 neutralizing monoclonal antibodies (mAbs), named 2H12 and 8F12, isolated from antigen immunized mouse hybridomas. As there is no vaccine or effective therapeutic treatment of the virus, the described two mAbs showed potent efficacy targeting two clades (clade B and clade D) of EV-D68 strain when used in combination. Two mAbs showed significant survival improvement using mouse infection model and neutralized two different clades of EV-D68 strain with different potencies in cell culture assay. Cryo-EM structure analysis of the Fab complex with EV-D68 showed a similar epitope binding region for both antibodies.

Q3-1: Although the two antibody cocktail (1:1 ratio) had differential neutralizing activity against two different clades in cocktail, the combined treatment data did not demonstrate synergistic activity. Therefore, the proposed usage of cocktail of two mAbs is not well supported by the data set and there is a limited therapeutic value in consideration of more costly for development of cocktail in comparison with development of single monoclonal antibody.

A3-1: We agree with the reviewer that the combined treatment data did not demonstrate synergistic activity. However, we need to emphasize the fact that the 2H12/8F12 cocktail conferred broader neutralization and *in vivo* protection than single MAbs (Fig. 1a and 2), supporting the notion that use of antibody cocktail is a better option for neutralizing and treating diverse EV-D68 clades than the use of a single antibody.

Antibody cocktail has been used in the treatment of viral infections. For example, Regeneron's antibody cocktail REGN-EB3 (Inmazeb®) is a FDA-approved treatment for Ebola and please note that REGN-EB3 is a cocktail of three monoclonal antibodies, REGN3470, 3471, and 3479. In addition, the two antibodies (2H12 and 8F12) could be made into a bispecific antibody, the cost of which is comparable to the cost of a

single antibody.

Q3-2: Major points to be addressed:

- Title: The study did not cover 'development' aspect, should take out 'Development' out the title.

A3-2: Thanks for the comment. As suggested, the title has been changed to "Functional and structural characterization of a two-MAb cocktail for delayed treatment of enterovirus D68 infections".

Q3-3: Rational of using cocktail of two mAbs in the study is not well defined. Authors should run a binding competition assay to dissect whether the two mAbs can compete each other in viral binding.

A3-3: Rationale for using cocktail of two mAbs was mentioned in our study, to read "The distinct cross-neutralization profiles of 2H12 and 8F12 propelled us to investigate whether they can be used in combination" (please lines 159-160). MAb 2H12 and 8F12 displayed distinct cross-neutralization profiles. Specially, 2H12 potently neutralized the clade D strain 18953 but almost had no neutralization effect on the prototype strain Fermon whereas 8F12 efficiently neutralized Fermon but poorly neutralized strain 18953 (Fig. 1a). By contrast, the 2H12/8F12 cocktail potently neutralized all tested EV-D68 strains (Fig. 1a). Thus, compared with single MAbs, antibody cocktail showed broader neutralizing effects. To emphasize this point, we have modified the statements in our revised manuscript (please see lines 159-160, 166-168), to read "The distinct cross-neutralization profiles of 2H12 and 8F12 propelled us to investigate whether they can be used in combination **to improve neutralization breadth.**" and the conclusion "In another word, neutralization breadth was improved by combining the 2H12 and 8F12 antibodies, providing a strong rationale for using the cocktail but not single MAbs for pan-EV-D68 neutralization".

As suggested, a competitive binding assay using BLI was performed in order to examine whether MAb 2H12 would compete with MAb 8F12 for viral binding. In this experiment, immobilized EV-D68 virion was saturated with the first antibody 2H12 and then allowed to bind to the second antibody 8F12. As shown in the new Fig. 6e, the antigen binding of 2H12 blocked the binding of 8F12, suggesting that the two MAbs bind to similar or overlapping epitopes. The results are consistent with the structural observation that the 2H12 and 8F12 MAbs have very similar footprints on the virion capsid surface (Fig. 4f and 5l).

Fig. 6. Comparison of the two immune complex structures. **e** A competitive binding assay using BLI. Immobilized EV-D68 virion (18947 strain) was saturated with the first antibody 2H12 and then allowed to react with 2H12 alone (control) or the antibody mixture (8F12 + 2H12).

Q3-4: Analysis of cryo-EM structures of the two viral/Fab complexes did not clearly point out the differences of the two mAbs in binding epitopes and the mechanisms of action of the two mAbs.

A3-4: Thanks for the comment. In general, the binding footprints of 2H12 and 8F12 on the virion surface are very similar except that 8F12 can also interact with the VP1 BC loop (Fig. 4f and 5l), suggesting that the two antibodies may target the same antigenic site. To verify this structure-based prediction, we performed a BLI competitive binding assay, in which immobilized EV-D68 virion was saturated with the first antibody 2H12 and then allowed to bind the second antibody 8F12. As shown in the Fig. 6e, initial binding of 2H12 to the EV-D68 virions blocked subsequent 8F12 binding. Hence, both structural and biochemical data demonstrate that the binding epitopes of 2H12 and 8F12 are similar or largely overlapping (Fig. 6f).

Both 2H12- and 8F12-binding footprints overlap with the sialic acid receptor binding site (Fig. 4d, 4f, 5j, 5l) and both MAbs can inhibit the binding of EV-D68 to sialic acid receptor in HI assays (Fig. 3), indicating that blockade of EV-D68 binding to sialic acid receptors via steric hindrance is the main neutralization mechanism shared by MAbs 2H12 and 8F12 (Fig. 6f). In addition, our structural data showed that 2H12 binding could destroy viral particles to some extent (Supplementary Fig. 6a) and trigger premature virus uncoating (Fig. 5g-i), likely leading to impaired virus infectivity. Therefore, MAb 2H12 may adopt two additional mechanisms to achieve neutralization, including impairing virion integrity and inducing premature virion uncoating (Fig. 6f).

To make our point clear, we have now added the summary of the epitopes and neutralization mechanisms to Fig. 6f and modified the manuscript accordingly (please see lines 409-426).

MAb	Epitope	Neutralization mechanism
2H12	VP1-GH loop	1. Blocking virus-receptor interactions
	VP2-EF loop	2. Impairing viral particles
	VP3-C terminus	3. Inducing premature virus uncoating
8F12	VP1-BC loop	1. Blocking virus-receptor interactions
	VP1-GH loop	
	VP2-EF loop	
	VP3-C terminus	

Fig. 6. Comparison of the two immune complex structures. f Summary of the epitopes and neutralization mechanisms of the 2H12 and 8F12 MAbs.

Q3-5: Need to clarify if the two antibodies work cooperatively or independently as the data suggest that two mAbs neutralize two different clades of EV-D68 and appears as additive in action.

A3-5: Our biochemical and structural data indicate that the 2H12 and 8F12 MAbs work independently but not cooperatively. Cross-neutralization assay indicated that the antibody combination (2H12/8F12) is no more effective than the most effective individual MAb (2H12 for strain 18953; 8F12 for strain Fermon) (Fig. 1a), suggesting that there is no synergy when the two MAbs were combined. To emphasize this point, we have modified the manuscript accordingly, to read “It is likely that 2H12 and 8F12 act independently and exert preferential neutralization on different virus strains, as the 2H12/8F12 combination was no more effective than the most effective individual MAb for a given strain (eg. 2H12 for strain 18953 and 8F12 for strain Fermon) (Fig. 1a).” (please see lines 473-476).

Q3-6: The study made chimeric (mouse variable/human constant region) version of the two antibodies in the study (Figure 7). Humanized antibodies need to use human germline in variable sequences and commonly do CDR grafting into human germline to reach more than >90% sequences as human origin.

A3-6: Thanks for pointing this out. As suggested, the word “humanized” has been replaced with the words “human-mouse chimeric” or “chimeric” in our revised manuscript.

Other points in the data analysis, interpretation and conclusions:

Q3-7: Figure 1a: Binding affinity data (KD, kon, kdis) should follow after the Octet sensorgrams for easy following and clarity. The data in Figure1a should have standard deviations calculated from minimum of three independent experimental repeats.

A3-7: Thanks for the suggestions. We have put together the binding affinity data and

the Octet sensorgrams (please see the new Fig. 1d-e). In addition, the cross-neutralization experiments were repeated several times, and the neutralization concentration values of the MABs were the same.

Fig. 1 Neutralization activity and binding properties of the MABs. **d, e** Binding affinities of the MABs to EV-D68 18947 virion measured by BLI. MAB concentrations used and values of KD, Kon and Kdis were shown.

Q3-8: Figure 7b: It is misleading and not conclusive to demonstrate humanization using WB detection with anti-mouse IgG or anti-human IgG. Purity of antibodies and Fab used in structure work should be analyzed using size exclusion (SEC) chromatography to determine other contaminants, aggregates, and purity.

A3-8: Thanks for the comments. Anti-human IgG or anti-mouse IgG secondary antibodies (HRP conjugate) used in this study recognize the constant domains of both heavy and light chains. Chimeric MABs 2H12 (c2H12) and 8F12 (c8F12) reacted with anti-human IgG (constant domains) antibody but not anti-mouse IgG (constant domains) antibody in western blot assays (Fig. 7b), confirming their chimeric nature. In addition, sequencing analysis of the plasmids used for chimeric antibody expression confirmed that the variable regions of the antibody are of murine origin and the constant regions are of human origin. These data are sufficient to demonstrate chimerization of the antibodies.

As suggested, we use size exclusion chromatography to determine the purity of the recombinant chimeric MABs. We have added this result and modified the manuscript accordingly, to read “The purified c2H12 and c8F12 were > 94% purity as determined by size exclusion chromatography (Supplementary Figure 9)”

Supplementary Figure 9. The purity of the recombinant MAbs c2H12 (a) and c8F12 (b) was determined by size exclusion chromatography. The retention time and area of all the elution peaks are shown in the inset. The third elution peak of c2H12 and the second peak of c8F12 represent antibody monomer, whereas the first and second peaks of c2H12 and the first peak of c8F12 account for antibody aggregate and/or dimer.

References

1. Xiao C, Rossmann MG. Interpretation of electron density with stereographic roadmap projections. *J Struct Biol* **158**, 182-187 (2007).
2. Zupa E, *et al.* The cryo-EM structure of a gamma-TuSC elucidates architecture and regulation of minimal microtubule nucleation systems. *Nat Commun* **11**, 5705 (2020).
3. Liu Y, Sheng J, van Vliet ALW, Buda G, van Kuppeveld FJM, Rossmann MG. Molecular basis for the acid-initiated uncoating of human enterovirus D68. *Proc Natl Acad Sci U S A* **115**, E12209-E12217 (2018).